# Graph-driven Autonomous Adaptation for Multi-stream Concept Drift

## Abstract

Multi-stream concept drift introduces substantial challenges beyond traditional single-stream scenarios, as inter-stream dependencies produce complex, evolving dynamics that place increasingly stringent demands on real-world forecasting tasks. Existing adaptation methods typically address drift in isolation, overlooking spatio-temporal correlations—not only between streams but also among drift events—and failing to enable synchronized adaptation across large-scale data streams. Furthermore, current graph-based approaches often rely on static, predefined embeddings, limiting adaptability in highly dynamic environments. To address these limitations, we propose GAMAD, a graph-driven autonomous adaptation framework for multi-stream concept drift that integrates spatio-temporal graph construction with dynamic and predictive adaptation mechanisms for multi-stream forecasting. GAMAD dynamically constructs correlation graphs from historical distributional statistics and subgraph structures, eliminating dependence on pre-defined topology and enabling generalizable representations. For online multi-stream evolution, it performs noise-tolerant windowing for accurate node-level drift detection, and then expands from drift-centric nodes to localized subgraphs based on current multi-stream correlations. To further enhance forecasting generalization, we employ a hierarchical topological matching strategy to retrieve and reuse previously observed drift patterns, enabling more predictable adaptation to inherently unpredictable drifts. Extensive experiments on three large-scale real-world datasets demonstrate that GAMAD consistently outperforms state-of-the-art baselines in forecasting performance. We also show applicability to recommendation scenarios, where continuous adaptation to evolving user preferences is essential. We release code at: https://anonymous.4open.science/r/GAMAD-6AAB.

## 1 Introduction

Concept drift is a long-standing challenge in non-stationary data stream environments, where the underlying data distributions evolve unpredictably over time Lu et al. (2018). This issue is prevalent in real-world applications such as social media analysis, traffic forecasting, and online recommendation systems Jain et al. (2022). As data streams rapidly evolve, traditional models trained on static distributions struggle to accommodate new samples, making static learning paradigms inadequate Liu et al. (2023). Adaptive learning, which transforms static models into dynamic ones, has therefore become a prevailing solution to this problem Gama et al. (2014).

A rich line of research has explored adaptive strategies based on conventional machine learning models. These strategies are commonly categorized into active or passive approaches depending on whether the model initiates updates or responds reactively Halstead et al. (2023). Most existing studies focus on single-stream drift, using techniques such as ensembles, periodic retraining, and dynamic embeddings to adapt to new distributions Dong et al. (2017). Although effective in controlled settings, single-stream adaptation is fundamentally limited in real-world multi-stream environments such as urban sensor networks, regional weather systems, and multi-topic online platforms Gao et al. (2022).

The challenge in multi-stream settings stems not only from increased data volume but also from dynamic—often localized—correlations among streams. Simply replicating single-stream strategies

across multiple streams proves ineffective Yu et al. (2023). As streams co-evolve, their correlations may shift dramatically, especially during localized drift events Zhou et al. (2023). For instance, forest fires or traffic congestion can induce heterogeneous and temporally misaligned drift patterns across distributed sensor streams. As illustrated in Fig. 1, a sudden forest fire can trigger immediate drift in nearby sensors (A and B), while distant sensors (e.g., C) exhibit delayed responses due to the propagation of the event.

Recent studies have begun to address multi-stream concept drift, yet notable limitations remain. Some approaches transfer drift knowledge across limited source–target stream pairs, yielding incremental gains over single-stream baselines Chandra et al. (2016) but relying on strong assumptions that hinder generalization. Others are tailored to specific drift types and degrade under complex or irregular dynamics Zhou et al. (2023). Although recent advances leverage graph structures to capture stream dependencies, they typically emphasize historical pattern extraction and depend heavily on fine-tuning with online samples Rahmani et al. (2023). Critically, dynamic correlation shifts between streams during adaptation are often overlooked.

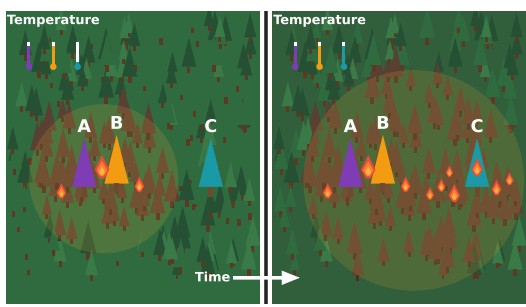

Figure 1: Illustration of stream-level correlation shift during a forest fire event. Sensors A and B, located near the ignition point, experience concurrent drift, while sensor C is affected with a delay as the fire propagates.

**Our key insight** is that effective adaptation to multi-stream concept drift requires a framework that not only captures evolving stream-level correlations but also localizes drift signals and leverages known patterns to generalize to unseen scenarios.

To this end, we propose GAMAD, a graph-driven autonomous adaptation framework for multi-stream concept drift. GAMAD constructs a dynamic correlation graph using dual-mode sampling from historical distributional statistics and subgraph patterns. The sampling evolves temporally from global sparsity to local density, enabling flexible representation of fine-grained and evolving inter-stream dependencies without relying on any pre-defined topology. For drift detection, we develop a node-level dynamic mechanism that adaptively sets thresholds based on each node's historical and current behavior. Once drifted nodes are identified, the model expands around them using real-time spatio-temporal correlations to construct localized subgraph-level drift regions. While concept drift remains inherently unpredictable, GAMAD enables forecasting generalization by retrieving and reusing known drift patterns via a hierarchical subgraph-to-node matching strategy, further enhanced by weighted sample augmentation for previously unseen drift.

The contributions of this work are summarized as follows:

- We propose GAMAD, a graph-driven autonomous adaptation framework for concept drift in evolving multi-stream environments. Leveraging dual-mode sampling over historical distributional statistics and subgraph structures, GAMAD constructs a dynamic correlation graph that evolves from global sparsity to local density, capturing fine-grained and evolving spatio-temporal dependencies without relying on pre-defined topologies.

- We develop a node-level dynamic drift detection mechanism with per-node adaptive thresholds. Upon detection, the model expands localized drift regions via real-time correlations and reuses previously observed patterns through a hierarchical subgraph-to-node matching strategy and weighted sample augmentation, making adaptation more predictable and generalizable to unseen drift.

- We conduct comprehensive experiments on three large-scale real-world multi-stream datasets, demonstrating that GAMAD consistently outperforms state-of-the-art baselines, particularly under irregular and previously unseen drifts with dynamic inter-stream interactions.

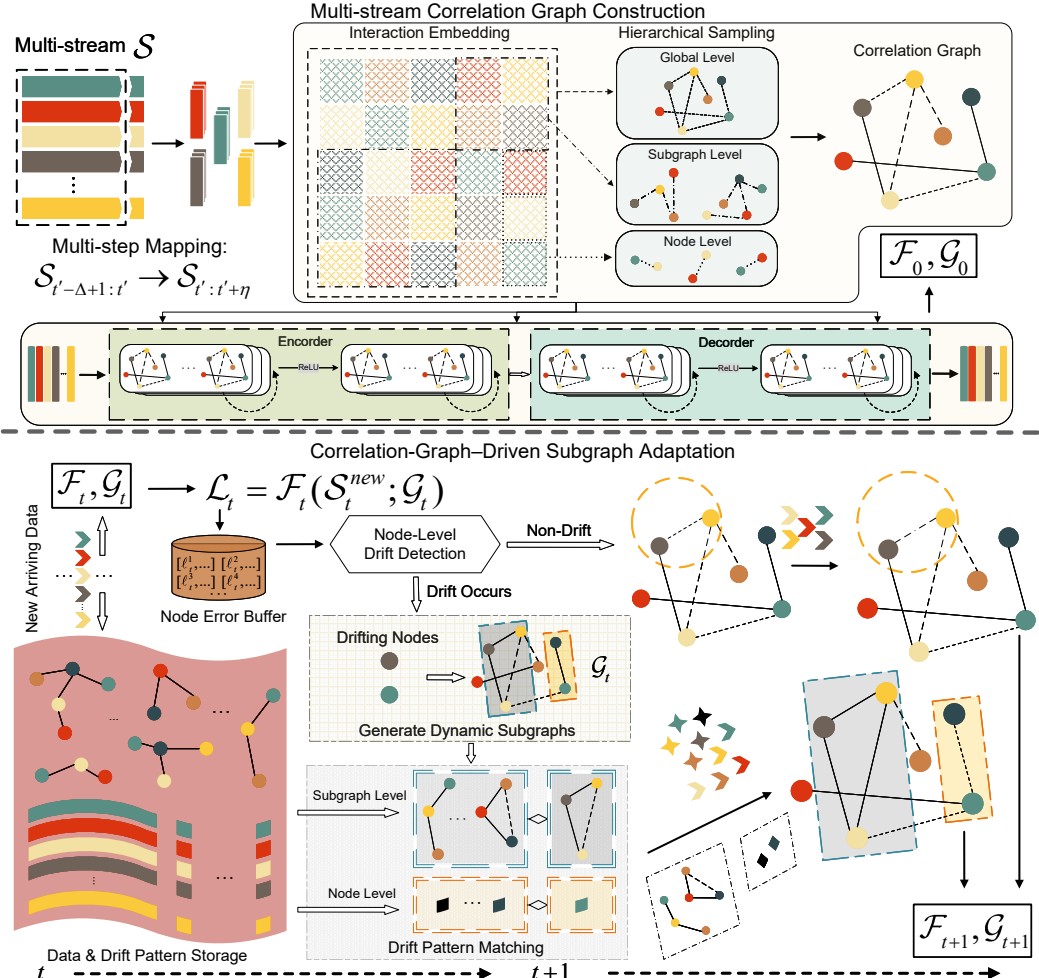

Figure 2: Overview of GAMAD. Top: starting from the multi-stream $\mathcal{S}$ (Stream nodes are represented by different colors), the framework builds a correlation graph $\mathcal{G}_0$ via interaction embeddings and Global–Subgraph–Node hierarchical sampling, and initializes a seq2seq DCGRU forecaster $\mathcal{F}_0$. Bottom: during online updates, node-level error buffers detect drift and the current graph guides expansion of drift-centric dynamic subgraphs; matched historical patterns are reused via subgraph-to-node matching to augment samples. Finally, the forecaster and the graph are locally updated to produce $\mathcal{F}_{t+1}$ and $\mathcal{G}_{t+1}$ (conservative updates under non-drift), realizing graph-driven autonomous adaptation.

## 2 METHODOLOGY

In this section, we present a detailed description of the proposed GAMAD framework. An overview of its architecture is shown in Fig. 2. Designed for multi-stream concept drift adaptation, the framework leverages a graph structure to guide both model initialization and the adaptation process during data stream updates.

### 2.1 PROBLEM SETUP AND NOTATION

**Multi-stream.** We consider a multi-stream $\mathcal{S} = \{\mathcal{D}^i\}_{i=1}^N$, where each stream $\mathcal{D}^i$ contains the same set of $d$ feature time series shared across all streams. Let $\mathbf{x}_t^i \in \mathbb{R}^d$ be the feature vector of stream $i$ at time $t$, and define $\mathcal{S}_t = [\mathbf{x}_t^1; \ldots; \mathbf{x}_t^N] \in \mathbb{R}^{N \times d}$. Throughout this paper, we focus on the subset of feature time series that are common to all streams; for example, a city's weather stations constitute a multi-stream, and we consider the same measurement (e.g., wind speed) across stations.

**Correlation graph.** Inter-stream dependencies are represented by a (potentially time-varying) correlation graph $\mathcal{G} \in \mathbb{R}^{N \times N}$ with adjacency $A = \text{Adj}(\mathcal{G})$. $\mathcal{G}$ captures both fixed factors (e.g., spatial layout in traffic networks) and dynamic factors from interactions among streams. In practice, global structure is relatively stable, while local correlations can vary over time, especially near drift events. We denote by $\mathcal{G}_{t-1}$ the graph available at step $t$ (its adjacency $A_{t-1} = \text{Adj}(\mathcal{G}_{t-1})$ will be used later in Section 2.4).

**Concept drift in multi-streams.** Let $\mathcal{P}_t(\mathcal{S})$ denote the data-generating distribution at time $t$. Concept drift occurs at $t+1$ if $\mathcal{P}_{t+1}(\mathcal{S}) \neq \mathcal{P}_t(\mathcal{S})$, and the multi-stream is said to drift if at least one stream $\mathcal{D}^i$ changes distribution. Drift correlations across streams can be *concurrent* (nearly simultaneous) or *delayed* (propagating with lags).

**Forecasting task.** Given an observed $\Delta$-step window and $\mathcal{G}_{t-1}$, the model predicts the next $\eta$ steps:

$$\mathcal{S}_{t+1:t+\eta} = f_\theta(\mathcal{S}_{t-\Delta+1:t}, \mathcal{G}_{t-1}). \tag{1}$$

Let $\ell(\cdot)$ be the multi-step forecasting loss. Training solves

$$\theta^\star, \mathcal{G}_t^\star = \arg\min_{\theta, \mathcal{G}_t} \ell(f_\theta(\mathcal{S}_{t-\Delta+1:t}, \mathcal{G}_{t-1}), \mathcal{S}_{t+1:t+\eta}), \tag{2}$$

where $f_\theta \in \mathcal{F}$ is parameterized by $\theta$. Unlike standard forecasting, we explicitly handle concept drift during testing: our framework (i) constructs and updates the multi-stream correlation graph $\mathcal{G}$, (ii) initializes the predictor $f_\theta$, and (iii) performs graph-driven online adaptation.

## 2.2 MULTI-STREAM CORRELATION GRAPH CONSTRUCTION

Pre-defined graphs are constrained by limited side information and rigid priors. Given the evolving nature of data streams, the correlation graph should capture complex spatio-temporal dependencies from historical data while retaining strong generalization to unseen drift patterns.

Starting from the historical multi-stream $\mathcal{S}^{\text{tr}}$, we construct a pairwise interaction matrix that encodes synchronous and asynchronous co-evolution between stream nodes. This transforms raw temporal signals into a relation-aware space, allowing edge weights to be estimated with improved accuracy and robustness. Formally, each $\mathcal{D}^i$ is encoded by a temporal encoder $\mathcal{E}(\cdot)$ into a feature vector $\mathbf{h}_i \in \mathbb{R}^d$. For every node pair $(\mathcal{D}^u, \mathcal{D}^v)$, we form an interaction embedding $\mathbf{p}_{uv} = \mathbf{h}_u \oplus \mathbf{h}_v$, where $\oplus$ denotes concatenation, and compute an unnormalized connection score $\psi_{uv} = g_\phi(\mathbf{p}_{uv})$ via a link predictor $g_\phi(\cdot)$. The resulting matrix $\Psi = [\psi_{uv}] \in \mathbb{R}^{N \times N}$ provides the foundation for a "Global–Subgraph–Node" hierarchical sampling process that progressively refines inter-stream dependencies.

**Global level (exploratory).** We aim to explore a broad range of potential inter-stream dependencies while keeping graph construction differentiable. The interaction score $\psi_{uv}^{(\text{G})}$ is taken directly from $\Psi$. We inject a Gumbel perturbation for each pair $(u,v)$, $\gamma_{uv} = -\ln(-\ln(\rho_{uv}))$ with $\rho_{uv} \sim \mathcal{U}(0,1)$, and obtain a stochastic, row-normalized weight

$$\mathcal{A}_{uv}^{\text{Gum}} = \frac{\exp((\psi_{uv}^{(\text{G})} + \gamma_{uv})/\kappa)}{\sum_{k \neq u} \exp((\psi_{uk}^{(\text{G})} + \gamma_{uk})/\kappa)},$$

where $\kappa$ is a temperature parameter shared across sampling modules and gradually annealed during training.

**Subgraph level (sparse and stable).** To promote sparsity and salient local connections, we apply differentiable subset sampling (DSS) to a pre-processed score matrix $\psi_{uv}^{(\text{D})}$ derived from $\Psi$. For each source node $u$, logits are normalized to zero mean and unit variance:

$$\tilde{\psi}_{uv}^{(\text{D})} = \frac{\psi_{uv}^{(\text{D})} - \mu_{\text{row}(u)}}{\sigma_{\text{row}(u)} + \epsilon}, \qquad \mathcal{A}_{uv}^{\text{DSS}} = \frac{\exp(\tilde{\psi}_{uv}^{(\text{D})}/\kappa)}{\sum_{k \neq u} \exp(\tilde{\psi}_{uk}^{(\text{D})}/\kappa)}.$$

**Node level (adaptive prior).** To capture long-term structural dependencies that may not be evident from short-term interactions, we learn two node-embedding dictionaries $E_1, E_2 \in \mathbb{R}^{N \times d}$ and compute an adaptive affinity

$$\psi_{uv}^{(\mathrm{A})} = \mathrm{ReLU}\big(E_1[u, :] \cdot E_2[v, :]\big), \qquad \mathcal{A}_{uv}^{\mathrm{ADP}} = \frac{\exp\big(\psi_{uv}^{(\mathrm{A})}\big)}{\sum_{k \neq u} \exp\big(\psi_{uk}^{(\mathrm{A})}\big)}.$$

**Fusion before diffusion.** After constructing $\mathcal{A}^{\mathrm{Gum}}$, $\mathcal{A}^{\mathrm{DSS}}$, and $\mathcal{A}^{\mathrm{ADP}}$, we integrate them into a unified correlation graph prior to diffusion convolution:

$$\mathcal{G} = \sum_{p=0}^{P} \Big(\mathcal{A}^{\mathrm{Gum}} \mathcal{W}_{p1} + \mathcal{A}^{\mathrm{DSS}} \mathcal{W}_{p2} + \mathcal{A}^{\mathrm{ADP}} \mathcal{W}_{p3}\Big), \tag{3}$$

where $P$ denotes the number of diffusion steps and $\mathcal{W}_{p*}$ are learnable coefficients for the $p$-th order.

## 2.3 DIFFUSION GRAPH CONVOLUTIONAL NETWORK

To capture temporal dependencies in multi-stream forecasting, we employ the Diffusion Convolutional Gated Recurrent Unit (DCGRU) Li et al. (2018), which ensures stable convergence under incremental updates. Diffusion convolution aligns with our adaptive correlation graph $\mathcal{G}$ by propagating correlations across neighboring nodes while preserving structural priors, thus improving generalization to drift patterns.

Built on a seq2seq framework, DCGRU realizes the multi-step mapping $\mathcal{S}_{t-\Delta+1:t} \mapsto \hat{\mathcal{S}}_{t+1:t+\eta}$, where the encoder consumes the input window to produce a final hidden state and the decoder rolls it out for $\eta$ steps. Formally, given $\mathcal{G}$ and an input $Z \in \mathbb{R}^{N \times d}$ (e.g., $Z = [\mathcal{S}_{t^*} \| h_{t^*-1}]$), diffusion convolution is defined as

$$(\Theta_\lambda \star_{\mathcal{G}} Z) = \sum_{p=0}^{P} \Big(\theta_{p,1}^{\lambda}(M_O^{-1}\mathcal{G})^p + \theta_{p,2}^{\lambda}(M_I^{-1}\mathcal{G}^\top)^p\Big) Z,$$

where $M_O$ and $M_I$ are the out-degree and in-degree diagonal matrices, and $\theta_{p,\cdot}^{\lambda}$ are learnable parameters for filter $\Theta_\lambda$, with $\lambda \in \{r, z, c\}$ denoting reset, update, and candidate gates. The recurrent updates are

$$r_{t^*} = \sigma\big(\Theta_r \star_{\mathcal{G}} [\mathcal{S}_{t^*} \| h_{t^*-1}] + b_r\big), \qquad\qquad z_{t^*} = \sigma\big(\Theta_z \star_{\mathcal{G}} [\mathcal{S}_{t^*} \| h_{t^*-1}] + b_z\big),$$

$$\tilde{h}_{t^*} = \tanh\big(\Theta_c \star_{\mathcal{G}} [\mathcal{S}_{t^*} \| (r_{t^*} \odot h_{t^*-1})] + b_c\big), \qquad h_{t^*} = z_{t^*} \odot h_{t^*-1} + (1 - z_{t^*}) \odot \tilde{h}_{t^*},$$

where $\sigma(\cdot)$ denotes the sigmoid function and $\odot$ element-wise multiplication.

To train the forecasting model, we employ Mean Absolute Error (MAE) averaged over all forecasting steps, streams, and features:

$$\mathcal{L} = \frac{1}{\eta N d} \sum_{t=1}^{\eta} \sum_{i=1}^{N} \sum_{j=1}^{d} \big|\hat{\mathcal{S}}_{ijt} - \mathcal{S}_{ijt}\big|,$$

where $\mathcal{S} \in \mathbb{R}^{\eta \times N \times d}$ denotes the ground truth and $\hat{\mathcal{S}} \in \mathbb{R}^{\eta \times N \times d}$ the predictions.

## 2.4 GRAPH-DRIVEN ONLINE ADAPTATION

While the DCGRU-based backbone provides a stable sequence-to-sequence mapping under stationary conditions, real-world multi-streams inevitably undergo concept drift during deployment. We therefore perform *graph-driven online adaptation* at inference time, leveraging the correlation graph $\mathcal{G}$ at each step as the dynamic basis for localized updates and knowledge reuse.

**Streaming protocol.** At time step $t$, the multi-stream input $\mathcal{S}_t$ arrives sequentially in a one-by-one manner, forming an online update slice $\mathcal{S}_t^{\mathrm{upd}}$. A history repository $\{\mathcal{S}^{\mathrm{hist}}\}$ accumulates such slices over time. Let $A_{t-1} = \mathrm{Adj}(\mathcal{G}_{t-1}) \in \mathbb{R}^{N \times N}$ denote the current adjacency derived from $\mathcal{G}_{t-1}$, which will be updated to $\mathcal{G}_t$ after assimilating the new slice.

**Node-level drift detection.** We monitor errors at the level of individual stream nodes. For each node $i$, maintain a buffer $B_i = \{e_{t-W+1}(i), \ldots, e_t(i)\}$ of length $W$, where $e_t(i)$ is the mean absolute error at time $t$. A dynamic threshold is estimated as $\tau_t(i) = \mathrm{median}(B_i) + \beta \cdot \mathrm{std}(B_i)$ with $\beta > 0$ (typically $\beta \approx 0.8$). A node is marked as drifted if its error persistently exceeds $\tau_t(i)$, and the set of drifted nodes is denoted by $\mathcal{I}_t$.

**Correlation-driven dynamic subgraphs.** Starting from $\mathcal{I}_t$, we expand subgraphs on $A$ within $h$ hops using both out-/in-neighbors and a top-$K$ rule under a quantile-based edge threshold. Let $\theta_q = \mathrm{Quantile}(A, q)$ with $q \in [q_{\min}, q_{\max}]$ (e.g., 95–100th), and define one-hop neighbors of $u$ by

$$\mathcal{N}_{\theta_q}(u) = \{v : A_{uv} > \theta_q\} \cup \{v : A_{vu} > \theta_q\}, \qquad \mathcal{N}_{\theta_q}^{\text{top-}K}(u) = \text{Top-K}_{\text{nbh}}(\mathcal{N}_{\theta_q}(u)). \tag{4}$$

Iterating $h$ hops from each $u \in \mathcal{I}_t$ yields a subgraph node set $\mathcal{V}_t^{(m)}$ per seed; we adjust $q$ via bisection so that $\left|\cup_m \mathcal{V}_t^{(m)}\right| \leq N_{\text{sg}}^{\max}$. Conditioning updates on $\mathcal{G}$ through such correlated subgraphs constrains adaptation to relevant regions, mitigating catastrophic forgetting while exploiting inter-stream dependencies.

**Subgraph-to-history matching (knowledge reuse).** For each subgraph $\mathcal{V}_t^{(m)}$, extract the corresponding slice $\mathcal{S}_t^{(m)} \subseteq \mathcal{S}_t^{\text{upd}}$. We match $\mathcal{S}_t^{(m)}$ against historical slices $\{\mathcal{S}^{\text{hist}}\}$ to identify recurrent or similar drifts using a hybrid strategy that combines subgraph-level and node-level similarities. At the subgraph level, define

$$d_{\text{sub}}(m, s) = \frac{\left\| \mathcal{S}_t^{(m)} - \Pi_{\mathcal{V}_t^{(m)}}(\mathcal{S}^{\text{hist}}[s]) \right\|_F}{\sqrt{\eta \cdot |\mathcal{V}_t^{(m)}|}}, \tag{5}$$

and accept a match if $\min_s d_{\text{sub}}(m, s) \leq \delta_{\text{sub}}$. If the subgraph-level criterion is not satisfied, we fall back to node-wise matching: let $\mathbf{v}_t^{(m)}(i)$ be the slice of node $i \in \mathcal{V}_t^{(m)}$ and $\mathbf{s}_s^{\text{hist}}(j)$ the node-$j$ slice in $\mathcal{S}^{\text{hist}}[s]$, then

$$d_{\text{node}}(i; s, j) = \frac{\left\| \mathbf{v}_t^{(m)}(i) - \mathbf{s}_s^{\text{hist}}(j) \right\|_2}{\sqrt{\eta}}, \qquad d_{\text{node}}^{\star}(i) = \min_{s,j} d_{\text{node}}(i; s, j). \tag{6}$$

A node is accepted if $d_{\text{node}}^{\star}(i) \leq \delta_{\text{node}}$, yielding the matched set $\widehat{\mathcal{V}}_t^{(m)} = \{ i \in \mathcal{V}_t^{(m)} : d_{\text{node}}^{\star}(i) \leq \delta_{\text{node}} \}$.

Given the top-$K$ matched historical patterns $\mathcal{M}^{(m)} = \{s_1, \ldots, s_K\}$ for the current drifted subgraph $\mathcal{V}_t^{(m)}$ (unmatched cases degenerate to micro-subgraphs with a single node), we update the new multi-stream slice based on the matched node histories. For each node $i \in \mathcal{V}_t^{(m)}$, assign descending weights $w_{1,i} > w_{2,i} > \cdots > w_{K,i}$ with $\sum_{k=1}^{K} w_{k,i} = 1$, and fuse as

$$\widetilde{\mathcal{S}}_t^{(m)}[i] = \sum_{k=1}^{K} w_{k,i} \, \Pi_i\big(\mathcal{S}^{\text{hist}}[s_k]\big), \quad i \in \widehat{\mathcal{V}}_t^{(m)}. \tag{7}$$

Embed $\widetilde{\mathcal{S}}^{(m)}$ into $\mathcal{S}_t^{\text{upd}}$ by batchwise concatenation to form $\widetilde{\mathcal{S}}_t^{\text{upd}}$, improving fit to recurrent/similar drifts and enhancing generalization by reusing multiple related patterns.

We summarize online adaptation with a unified objective, where the drift indicator $\delta_t$ controls both sample choice and the strength of parameter/graph updates:

$$\min_{\theta, \mathcal{G}} \ \ell\big(f_\theta(\mathcal{X}_t, \mathcal{G}_{t-1}), \mathcal{S}_{t+1:t+\eta}\big) + \lambda_\theta(\delta_t) \|\theta - \theta_{t-1}\|_2^2 + \lambda_{\mathcal{G}} \left\| W(M_t) \odot (\mathcal{G} - \mathcal{G}_{t-1}) \right\|_F^2, \tag{8}$$

where $\mathcal{X}_t = (1 - \delta_t) \mathcal{S}_{t-\Delta+1:t}^{\text{upd}} + \delta_t \widetilde{\mathcal{S}}_{t-\Delta+1:t}^{\text{upd}}$.

Here, $\delta_t = 1$ indicates a detected drift, in which case augmented slices $\widetilde{\mathcal{S}}^{\text{upd}}$ are used, $\lambda_\theta(\delta_t)$ is relaxed, and $W(M_t)$ emphasizes updates on the drift-affected subgraph; $\delta_t = 0$ denotes no drift, where original slices $\mathcal{S}^{\text{upd}}$ are used, parameter updates are strongly constrained, and the graph remains stable.

https://anonymous.4open.science/r/GAMAD-6AAB

## 3 EXPERIMENTS

In this section, we conduct extensive experiments to demonstrate that the proposed GAMAD outperforms a wide range of existing forecasting models, including traditional statistical models, conventional graph-based predictors, and graph-based approaches specifically designed for multi-stream drift.

### 3.1 SETUP

**Datasets.** Our experiments are conducted on three large-scale real-world benchmark datasets spanning two domains. METR-LA Li et al. (2018) and PEMS-BAY Chavhan & Venkataram (2020) are widely used traffic datasets collected from highway sensor networks in Los Angeles County and the Bay Area, respectively. WEATHER Liu (2019) is a meteorological dataset from Beijing, China, where we focus on two representative features: surface air pressure (*psur*), which exhibits strong regularity, and relative humidity at 2 meters (*rh2m*), which shows larger short-term fluctuations. We follow the same data configurations as the reference works for fair comparison.

**Baselines.** We compare GAMAD against a wide range of representative baselines. For traditional regression methods, we include Historical Average (HA), Vector Auto-Regression (VAR), Support Vector Regression (SVR), and Fully Connected LSTM (FC-LSTM). Beyond these, we consider graph-based spatio-temporal models and recent advances in online adaptation, including DCRNN Li et al. (2018), Graph WaveNet Wu et al. (2019), AGCRN Bai et al. (2020), STFGNN Li & Zhu (2021), GTS Shang et al. (2021), MegaCRN Jiang et al. (2023), and CGLM Zhou & Lu (2025). Together, these methods span diffusion and adaptive convolutions, attention-based fusion, dynamic and discrete graph construction, as well as continuous graph learning for online adaptation.

**Evaluation Metrics.** We evaluate all methods using three standard metrics: mean absolute error (MAE), root mean square error (RMSE), and mean absolute percentage error (MAPE). Detailed descriptions of the datasets, experimental setup, and the formal expressions of the evaluation metrics are provided in Appendices C and D.

### 3.2 RESULTS

**Multi-Step Forecasting Results.** We report multi-step forecasting results on both traffic and weather benchmarks to assess accuracy, robustness, and cross-domain generalization. Traffic results on the canonical datasets (METR-LA and PEMS-BAY) establish performance on widely used benchmarks, while the WEATHER results complement the study by examining non-traffic dynamics with two representative meteorological features. We first present the traffic results and analysis, followed by the weather results and analysis.

As shown in Table 1, our proposed GAMAD consistently outperforms all baselines on both traffic benchmarks. Compared with the strongest traditional GNN baseline (MegaCRN), GAMAD reduces MAE by 6.75%, 6.20%, and 9.45% on METR-LA, and by 3.13%, 2.34%, and 4.69% on PEMS-BAY for short-, medium-, and long-horizon predictions, respectively, yielding an overall average improvement of 6.10%. Against the online baseline CGLM, which incorporates continuous graph learning, our method further improves MAE by 1.67%, 3.20%, and 5.74% on METR-LA and by 3.88%, 2.34%, and 4.69% on PEMS-BAY across the three horizons.

These improvements are closely tied to the characteristics of the datasets. On METR-LA, urban traffic is highly dynamic, with localized congestion and short-term distributional shifts that static GNNs fail to capture. Although CGLM introduces online updates, its reactive nature limits sensitivity to abrupt changes. By contrast, our drift-aware mechanism achieves more fine-grained adaptation, explaining the larger relative gains observed on this dataset. On PEMS-BAY, traffic patterns are generally smoother and drifts manifest more at the regional scale. While the improvements are relatively smaller, they remain consistent across horizons, demonstrating the robustness and generalizability of our approach to settings with less abrupt but still evolving dynamics.

*Forcasting evaluation results on WEATHER data sets are deferred to Appendix E.*

*Statistical significance tests validating the performance gains are provided in Appendix H.*

Table 1: Multi-Step Forecasting Performance Comparison (Traffic)

| | H | Metric | HA | VAR | FC-LSTM | DCRNN | Graph WaveNet | AGCRN | GTS | MegaCRN | CGLM | **GAMAD** |
|---|---|---|---|---|---|---|---|---|---|---|---|---|
| **METR-LA** | H-3 | MAE | 4.16 | 4.42 | 3.44 | 2.77 | 2.69 | 2.87 | 2.64 | 2.52 | 2.39 | **2.35** |
| | | RMSE | 7.80 | 7.89 | 6.30 | 5.38 | 5.15 | 5.58 | 4.95 | 4.94 | 4.28 | **4.18** |
| | | MAPE | 13.00% | 10.20% | 9.60% | 7.30% | 6.90% | 7.70% | 6.80% | 6.44% | 6.14% | **6.00%** |
| | H-6 | MAE | 4.16 | 5.41 | 3.77 | 3.15 | 3.07 | 3.23 | 3.01 | 2.93 | 2.71 | **2.66** |
| | | RMSE | 7.80 | 9.13 | 7.23 | 6.45 | 6.22 | 6.58 | 5.85 | 6.06 | 5.03 | **4.92** |
| | | MAPE | 13.00% | 12.70% | 10.90% | 8.80% | 8.37% | 9.00% | 8.20% | 7.96% | 7.38% | **7.24%** |
| | H-12 | MAE | 4.16 | 6.52 | 4.49 | 3.60 | 3.53 | 3.62 | 3.41 | 3.38 | 3.09 | **3.03** |
| | | RMSE | 7.80 | 10.11 | 8.69 | 7.59 | 7.37 | 7.51 | 6.74 | 7.23 | 5.82 | **5.70** |
| | | MAPE | 13.00% | 15.80% | 13.20% | 10.50% | 10.10% | 10.38% | 9.90% | 9.72% | 8.85% | **8.66%** |
| **PEMS-BAY** | H-3 | MAE | 2.88 | 1.74 | 2.05 | 1.38 | 1.30 | 1.37 | 1.32 | 1.28 | 1.29 | **1.27** |
| | | RMSE | 5.59 | 3.16 | 4.19 | 2.95 | 2.74 | 2.87 | 2.62 | 2.72 | 2.42 | **2.38** |
| | | MAPE | 6.80% | 3.60% | 4.80% | 2.90% | 2.73% | 2.94% | 2.80% | 2.67% | 2.71% | **2.64%** |
| | H-6 | MAE | 2.88 | 2.32 | 2.20 | 1.74 | 1.63 | 1.69 | 1.64 | 1.60 | 1.58 | **1.55** |
| | | RMSE | 5.59 | 4.25 | 4.55 | 3.97 | 3.70 | 3.85 | 3.41 | 3.68 | 3.10 | **3.03** |
| | | MAPE | 6.80% | 5.00% | 5.20% | 3.90% | 3.67% | 3.87% | 3.60% | 3.57% | 3.54% | **3.45%** |
| | H-12 | MAE | 2.88 | 2.93 | 2.37 | 2.07 | 1.95 | 1.96 | 1.91 | 1.88 | 1.83 | **1.80** |
| | | RMSE | 5.59 | 5.44 | 4.96 | 4.74 | 4.52 | 4.54 | 3.97 | 4.42 | 3.61 | **3.54** |
| | | MAPE | 6.80% | 6.50% | 5.70% | 4.90% | 4.63% | 4.64% | 4.40% | 4.41% | 4.31% | **4.26%** |

*Noise robustness analysis evaluating the stability of GAMAD under Gaussian and spike perturbations is provided in Appendix G.*

**Effect of Graph-Driven Framework.** We evaluate the contribution of each component in our graph-driven design through two groups of ablation studies: (1) offline hierarchical graph construction, and (2) online drift-adaptation mechanisms. This separation aligns with the two-stage structure of GAMAD, enabling a clear examination of how global, subgraph-level, and node-level correlations—as well as drift-aware online updates—affect final performance.

*Offline hierarchical graph construction.* Table 2 (Offline Ablation) reports the offline-only ablation results, where no online updates are applied. The global adjacency $\mathcal{A}^{\text{Gum}}$ (abbrev. $\mathcal{A}^G$), the subgraph-level adjacency $\mathcal{A}^{\text{DSS}}$ (abbrev. $\mathcal{A}^D$), and the node-level adaptive adjacency $\mathcal{A}^{\text{ADP}}$ (abbrev. $\mathcal{A}^A$) each provide meaningful predictive gains. Their pairwise combinations—$\mathcal{A}^{\text{Gum}} + \mathcal{A}^{\text{DSS}}$ (abbrev. $\mathcal{A}^{GD}$) and $\mathcal{A}^{\text{DSS}} + \mathcal{A}^{\text{ADP}}$ (abbrev. $\mathcal{A}^{DA}$)—typically yield further improvements across metrics and horizons. The full fusion $\mathcal{A}^{\text{Gum}} + \mathcal{A}^{\text{DSS}} + \mathcal{A}^{\text{ADP}}$ (abbrev. $\mathcal{A}^{GDA}$; w/o ol) achieves the best results across all horizons. These results demonstrate that the hierarchical adjacency design is not redundant: the three components capture complementary correlation structures at different spatial resolutions, leading to a more expressive offline graph representation (Fig. 3).

*Online graph-driven adaptation mechanisms.* Table 2 (Online Ablation) further examines the contribution of each component in the online phase. Naive online learning (abbrev. $\mathcal{M}^{NL}$) applies uniform updates without drift awareness and therefore brings only modest performance gains. Introducing node-level drift detection (DD; abbrev. $\mathcal{M}^D$) substantially improves adaptation by selectively updating affected nodes. Enhancing DD with dynamic subgraph expansion (DD+Subgraph; abbrev. $\mathcal{M}^{DS}$) further boosts performance by capturing localized drift-propagation regions. Finally, the full online model—combining drift detection, subgraph expansion, and historical drift-pattern reuse (Full; abbrev. $\mathcal{M}^F$)—achieves the best performance across all horizons, outperforming naive online learning by 4.1–7.8% in MAE (Fig. 4). These results confirm that each online mechanism contributes uniquely and complementarily to stable and efficient adaptation under evolving drift.

*Hyperparameter sensitivity results are deferred to Appendix F.*

*Runtime efficiency and computational complexity analysis of GAMAD are provided in Appendix I.*

Table 2: Evaluation of Graph-Driven Framework Components on METR-LA

| | | **Offline Ablation** | | | | | | | **Online Ablation** | | | |
|---|---|---|---|---|---|---|---|---|---|---|---|---|
| H | Metric | $\mathcal{A}^{G}$ | $\mathcal{A}^{D}$ | $\mathcal{A}^{A}$ | $\mathcal{A}^{GD}$ | $\mathcal{A}^{DA}$ | w/o ol | H | Metric | $\mathcal{M}^{NL}$ | $\mathcal{M}^{D}$ | $\mathcal{M}^{DS}$ | $\mathcal{M}^{Full}$ |
| H-3 | MAE | 2.60 | 2.55 | 2.63 | 2.54 | 2.54 | **2.52** | H-3 | MAE | 2.50 | 2.41 | 2.36 | **2.35** |
| | RMSE | 4.89 | 4.80 | 4.95 | 4.72 | 4.77 | **4.72** | | RMSE | 4.64 | 4.30 | 4.20 | **4.18** |
| | MAPE | 6.69% | 6.52% | 6.72% | 6.63% | 6.50% | **6.46%** | | MAPE | 6.33% | 6.21% | 5.95% | **6.00%** |
| H-6 | MAE | 3.00 | 2.92 | 3.01 | 2.92 | 2.92 | **2.89** | H-6 | MAE | 2.87 | 2.77 | 2.69 | **2.66** |
| | RMSE | 5.89 | 5.72 | 5.91 | 5.62 | 5.70 | **5.62** | | RMSE | 5.55 | 5.14 | 4.96 | **4.92** |
| | MAPE | 8.07% | 7.96% | 8.08% | 8.05% | 7.95% | **7.84%** | | MAPE | 7.73% | 7.55% | 7.24% | **7.24%** |
| H-12 | MAE | 3.41 | 3.31 | 3.42 | 3.34 | 3.33 | **3.27** | H-12 | MAE | 3.24 | 3.19 | 3.07 | **3.03** |
| | RMSE | 6.81 | 6.62 | 6.82 | 6.51 | 6.60 | **6.50** | | RMSE | 6.43 | 5.98 | 5.76 | **5.70** |
| | MAPE | 9.76% | 9.53% | 9.62% | 9.62% | 9.64% | **9.46%** | | MAPE | 9.32% | 9.10% | 8.83% | **8.66%** |

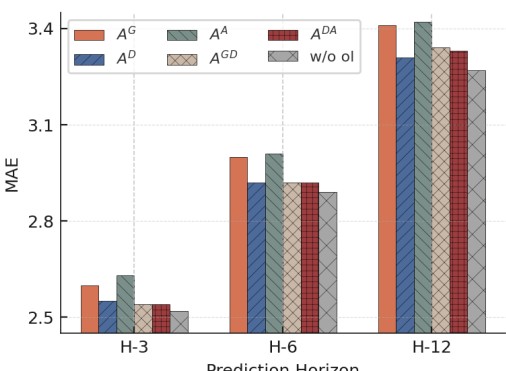

Figure 3: Offline Ablation MAE Comparison

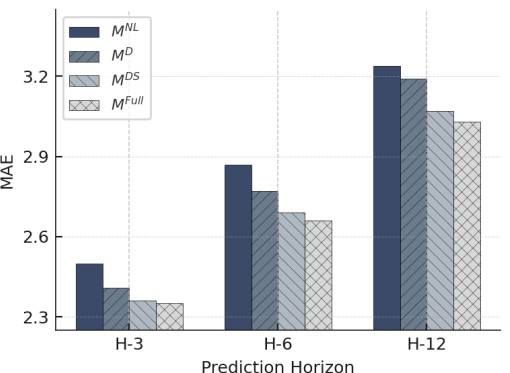

Figure 4: Online Ablation MAE Comparison

## 4 CONCLUSIONS

We presented GAMAD, a graph-driven autonomous adaptation framework for multi-stream concept drift that constructs a layered sampling graph to capture global and local dependencies, guides model initialization and online adaptation, and reuses drift patterns via hierarchical subgraph-to-node matching. Across traffic and weather benchmarks, GAMAD consistently outperforms state-of-the-art baselines in both offline and online settings, underscoring the benefits of correlation-guided, node-level adaptation. Going forward, we will target highly volatile multi-streams where within-stream distributional changes dominate and inter-stream correlations weaken; we will complement correlation-driven control with distribution- and uncertainty-centric signals when correlations are unreliable; and we will pursue fine-grained detection that distinguishes drift from noise under high variability.

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

# Appendix

## A  RELATED WORK

We briefly review prior work on concept drift adaptation and graph-based modelling, which form the foundation of our method. While effective in single-stream or static settings, existing approaches struggle to generalize to multi-stream drift scenarios with dynamic and correlated structures.

### A.1  CONCEPT DRIFT ADAPTATION STRATEGIES

Concept drift refers to changes in the underlying data distribution over time, which can significantly degrade model performance if not properly addressed Li et al. (2024b); Yang et al. (2025). Existing adaptation methods are broadly categorized into informed (detection-based) and blind (detection-free) approaches Agrahari & Singh (2022). Informed strategies rely on explicit detection mechanisms to trigger updates, typically based on metrics such as forecasting error Xu et al. (2020), distributional divergence Khezri et al. (2020), or statistical change detection within sliding windows Bifet & Gavalda (2007); Chiu & Minku (2020); Gözüaçık & Can (2021); Wang et al. (2020). Blind strategies, on the other hand, avoid detection and continuously update the model using streaming learners Wang et al. (2022), online pruning Gama et al. (2003), or adaptive ensemble methods Cano & Krawczyk (2022); Wen et al. (2024). These techniques have proven effective in single-stream environments, where data evolves in a relatively isolated and sequential manner.

In multi-stream scenarios, however, adaptation becomes considerably more complex due to interacting temporal dynamics and cross-stream dependencies. Many existing frameworks are restricted to two- or three-stream setups and adopt source–target designs akin to transfer learning Haque et al. (2017); Renchunzi & Pratama (2022), which limits scalability. Others focus on specific drift types or small-scale environments, reducing generalizability. More recent efforts leverage graph-based representations to capture inter-stream correlations Rahmani et al. (2023), yet they often rely on relatively simple online learning routines Zhou et al. (2023). These limitations highlight the need for frameworks that are both scalable and structure-aware, capable of localized adaptation across correlated streams while remaining robust to diverse drift dynamics.

### A.2  GRAPH NEURAL NETWORKS FOR MULTI-STREAM MODELLING

Graph Neural Networks (GNNs) have become powerful tools for modelling complex, structured data. By aggregating information from a node's neighbors, GNNs learn representations that capture both local and global topological features Li et al. (2024c), making them particularly suitable for multi-stream scenarios where correlations vary over time. Static GNNs, such as Graph Convolutional Networks (GCNs), have proven effective in applications like social networks and recommender systems Jiang & Luo (2022); Li et al. (2024a). However, many real-world systems exhibit dynamic behavior, where node relationships evolve continuously.

In traffic forecasting, a representative multi-stream application, models often construct correlation graphs using pre-defined topologies based on physical layouts or domain knowledge. For instance, DCRNN integrates such fixed graphs into a diffusion convolutional architecture Li et al. (2018). Graph-WaveNet Wu et al. (2019) and STFGNN Li & Zhu (2021) enhance this by introducing adaptive matrices to capture fine-grained dependencies. ASTGCN Guo et al. (2022) leverages Laplacian-based latent graphs, while DGCRN Li et al. (2023) uses learnable filters to generate data-driven dynamic graphs during training. Other notable models, including GMAN Zheng et al. (2020), STEP Shao et al. (2022a), and $D^2$STGNN Shao et al. (2022b), demonstrate that pre-training temporal graph structures can further improve performance in structured forecasting tasks.

To overcome the rigidity of fixed or heuristic structures, recent works have shifted toward data-driven graph learning. Adaptive matrix construction Bai et al. (2020) and sampling-based approaches such as DGSL Shang et al. (2021) infer evolving structures directly from data. More advanced designs include meta-graph-based models like MetaGCRN Jiang et al. (2023), which dynamically reconfigure

temporal graphs using time-aware node banks, and adaptive diffusion graph attention mechanisms Zhou & Lu (2025), which selectively update local subgraphs triggered by drift, balancing stability and adaptability.

### A.3 SUMMARY AND MOTIVATION

Together, prior studies on concept drift adaptation and graph-based modelling reveal both progress and limitations. Adaptation strategies have shown how models can remain responsive to evolving distributions, but their extension to multi-stream scenarios remains coarse and inefficient. At the same time, GNNs provide powerful tools for capturing dynamic correlations, yet existing approaches seldom integrate them with drift-aware mechanisms. These observations highlight an open opportunity: combining fine-grained drift detection with graph-driven structural adaptation to address the challenges of large-scale, correlated data streams. Our work builds directly on this gap, proposing an autonomous framework that unifies node-level drift signals and evolving graph structures to achieve scalable and reliable multi-stream adaptation.

## B NOTATIONS

For clarity, we summarize the key notations used in this paper in Table 3. We include symbols directly related to our framework across three aspects: (i) multi-stream inputs and correlation graph construction, (ii) drift detection and subgraph-to-history matching, and (iii) the unified online adaptation objective. Common RNN update variables (e.g., gate states) are omitted.

## C ADDITIONAL DETAILS OF DATA SETS

we evaluate GAMAD using three large-scale real-world benchmarks covering two representative multi-stream domains: traffic and weather. Table 4 reports their key statistics.

For the two traffic datasets (METR-LA and PEMS-BAY), we follow the standard practice and split the data into 70% training, 10% validation, and 20% testing. For the WEATHER dataset, which is relatively less complex and exhibits slower temporal dynamics, we use 50% of the data for training, 10% for validation, and the remaining 40% for testing. Such a setup ensures fair comparison, reproducibility, and consistency with prior work.

We briefly describe each dataset as follows:

- **METR-LA** (Los Angeles, USA) Li et al. (2018): Vehicle speed readings collected from 207 loop detectors on highways across Los Angeles County, sampled every 5 minutes over March to June 2012. The dataset reflects complex urban traffic with strong rush-hour periodicity and localized congestion events, where drifts often appear in specific segments or short time windows, posing challenges for fine-grained spatio-temporal adaptation.

- **PEMS-BAY** (Bay Area, USA) Chavhan & Venkataram (2020); Li et al. (2018): Traffic measurements collected by CalTrans' Performance Measurement System (PeMS) from 325 highway sensors, spanning January to May 2017 with 5-minute sampling. Compared with METR-LA, PEMS-BAY covers a larger region with more sensors, where traffic flows are relatively smoother and drifts often emerge at the regional scale rather than being confined to individual road segments. This makes it suitable for evaluating model robustness under broader and more aggregated drift patterns.

- **WEATHER** (Beijing, China) Liu (2019): Hourly meteorological records from 10 stations covering January 2015 to May 2018. We select two features: *psur* (surface air pressure), which follows highly regular seasonal cycles, and *rh2m* (relative humidity at 2 m), which shows substantial short-term fluctuations on top of regular trends. This contrast provides a complementary setting to traffic data, enabling evaluation of drift adaptation under both predictable and volatile dynamics.

Together, these datasets form a comprehensive testbed for evaluating drift adaptation under heterogeneous temporal dynamics.

Table 3: Notations used in the paper.

| Symbol | Type / Shape | Description |
|---|---|---|
| $\mathcal{S}$ | multi-stream | Multi-stream composed of data streams $\{\mathcal{D}^i\}_{i=1}^N$. |
| $\mathcal{S}_{t-\Delta+1:t}$ | $\mathbb{R}^{\Delta \times N \times d}$ | Input window of length $\Delta$ at time $t$. |
| $\mathcal{S}_{t+1:t+\eta}$ | $\mathbb{R}^{\eta \times N \times d}$ | Ground-truth target window of length $\eta$. |
| $\hat{\mathcal{S}}_{t+1:t+\eta}$ | $\mathbb{R}^{\eta \times N \times d}$ | Predicted target window. |
| $\mathcal{S}_t^{\mathrm{upd}}$ | slice | Online arriving slice at time $t$. |
| $\mathcal{S}^{\mathrm{hist}}$ | repository | Historical repository of slices for pattern matching. |
| $N, d, \Delta, \eta$ | scalars | Number of streams, feature dimension, input window length, and forecasting horizon. |
| $\mathcal{G}$ | $\mathbb{R}^{N \times N}$ | Correlation graph among streams. |
| $\mathcal{G}_{t-1}, \mathcal{G}_t$ | graphs | Graph before/after update at time $t$. |
| $A_{t-1}$ | $\mathbb{R}^{N \times N}$ | Adjacency matrix derived from $\mathcal{G}_{t-1}$, i.e., $A_{t-1} = \mathrm{Adj}(\mathcal{G}_{t-1})$. |
| $\Psi = [\psi_{uv}]$ | $\mathbb{R}^{N \times N}$ | Pairwise interaction score matrix. |
| $\mathcal{A}^{\mathrm{Gum}}, \mathcal{A}^{\mathrm{DSS}}, \mathcal{A}^{\mathrm{ADP}}$ | $\mathbb{R}^{N \times N}$ | Adjacency matrices from global (Gumbel), subgraph (DSS), and node-level (adaptive) sampling. |
| $E_1, E_2$ | $\mathbb{R}^{N \times d}$ | Node-embedding dictionaries used to compute $\mathcal{A}^{\mathrm{ADP}}$. |
| $\mathcal{W}_{p*}$ | weights | Fusion coefficients for $p$-th order prior to diffusion. |
| $P$ | scalar | Number of diffusion steps. |
| $\kappa$ | scalar | Shared temperature parameter for sampling. |
| $\mathcal{I}_t$ | set | Drifted node set detected at time $t$. |
| $\mathcal{V}_t^{(m)}$ | set | Expanded subgraph node set at time $t$ (seeded from $\mathcal{I}_t$). |
| $\widehat{\mathcal{V}}_t^{(m)}$ | set | Matched nodes within $\mathcal{V}_t^{(m)}$. |
| $\mathcal{M}^{(m)}$ | set | Top-$K$ matched historical patterns for subgraph $m$. |
| $h, K, q$ | scalars | Hop count, top-$K$, and quantile for thresholding. |
| $\theta_q$ | scalar | Edge threshold at quantile $q$, $\theta_q = \mathrm{Quantile}(A, q)$. |
| $\mathcal{N}_{\theta_q}(u)$ | set | One-hop neighbors of node $u$ above threshold $\theta_q$. |
| $\Pi_i, \Pi_{\mathcal{V}}$ | operators | Extract node-$i$ or node-set $\mathcal{V}$ trajectories from a slice. |
| $d_{\mathrm{sub}}(m, s)$ | scalar | Subgraph-level distance between current $\mathcal{S}_t^{(m)}$ and historical slice $s$. |
| $d_{\mathrm{node}}(i; s, j)$ | scalar | Node-level distance between node $i$ and historical node $j$. |
| $\delta_{\mathrm{sub}}, \delta_{\mathrm{node}}$ | scalars | Acceptance thresholds for subgraph/node matching. |
| $\widetilde{\mathcal{S}}_t^{(m)}$ | slice | Augmented slice for subgraph $m$ using matched histories. |
| $\widetilde{\mathcal{S}}_{t-\Delta+1:t}^{\mathrm{upd}}$ | slice | Batchwise concatenation of augmented slices. |
| $w_{k,i}$ | weights | Assigned weight of the $k$-th matched history for node $i$. |
| $\delta_t \in \{0, 1\}$ | indicator | Drift indicator at time $t$ (1: drift, 0: no drift). |
| $\mathcal{X}_t$ | slice | Input for adaptation: $(1 - \delta_t)\mathcal{S}_{t-\Delta+1:t}^{\mathrm{upd}} + \delta_t \widetilde{\mathcal{S}}_{t-\Delta+1:t}^{\mathrm{upd}}$. |
| $M_t$ | mask/index set | Drift-affected subgraph mask or index set at time $t$. |
| $W(M_t)$ | mask/weights | Graph update mask emphasizing drift-affected subgraphs. |
| $\lambda_\theta(\delta_t), \lambda_{\mathcal{G}}$ | scalars | Coefficients controlling parameter and graph updates. |
| $f_\theta$ | model | Forecasting model parameterized by $\theta$. |
| $\ell(\cdot)$ | loss | Training loss (MAE in our experiments). |

Table 4: Statistics of the benchmark datasets.

| Dataset | Stream Nodes | Samples | Update Rate |
|---|---|---|---|
| METR-LA | 207 | 34,272 | 5 min |
| PEMS-BAY | 325 | 521,160 | 5 min |
| WEATHER | 10 | 29,639 | 60 min |

## D  DETAILED EXPERIMENTAL SETTINGS

**Platform.** All models are implemented in PyTorch and executed on a high-performance server with NVIDIA A100 GPUs (80GB each) and Intel Xeon Gold CPUs.

**Initialization.** We adopt the hierarchical annealed sampling to construct the multi-stream correlation graph. The temperature is initialized at $\kappa_0 = 1.2$ and annealed to $\kappa_{\min} = 0.6$ for all datasets. The annealing interval is set to 3 on large-scale traffic datasets and 5 on the WEATHER dataset. The optimizer's initial learning rate is $\alpha_0 = 0.005$ for traffic datasets and $\alpha_0 = 0.001$ for WEATHER. Given that the hierarchical annealed sampling already yields highly generalizable connectivity patterns, we fix the diffusion order to $P = 1$ across all datasets.

**Graph-driven online adaptation.** We use a continuous node-level drift-detection window of size $W = 5$. When drift is detected, the dynamic subgraph around a drifted node is expanded up to $N_{\mathrm{sg}}^{\max} = 20$ nodes within $h$ hops following the correlation structure. For subgraph-to-history matching, we retrieve the top-$K = 3$ most similar subgraphs from both historical distributions and previously observed drift patterns. The similarity thresholds follow the method notation: $(\delta_{\mathrm{sub}}, \delta_{\mathrm{node}}) = (0.18, 0.20)$ for METR-LA and WEATHER, and $(\delta_{\mathrm{sub}}, \delta_{\mathrm{node}}) = (0.25, 0.30)$ for PEMS-BAY.

**Subgraph learning rates.** To update parameters on the drift-affected regions we use $\alpha_{\mathrm{dl}}$, and on non-drift timesteps we use $\alpha_{\mathrm{ol}}$. Specifically,

$$\alpha_{\mathrm{dl}} = \begin{cases} 5.0 \times 10^{-6}, & \text{METR-LA, WEATHER (\textit{psur})}, \\ 1.0 \times 10^{-5}, & \text{PEMS-BAY, WEATHER (\textit{rh2m})}. \end{cases}$$

$$\alpha_{\mathrm{ol}} = \begin{cases} 5.0 \times 10^{-6}, & \text{METR-LA, WEATHER}, \\ 1.0 \times 10^{-5}, & \text{PEMS-BAY}. \end{cases}$$

These settings align with the autonomous, node-level adaptation policy in Section. 2.4, enabling precise updates under drift while preserving stability elsewhere.

**Evaluation metrics.** We report three commonly used metrics for time series forecasting: mean absolute error (MAE), root mean square error (RMSE), and mean absolute percentage error (MAPE). Their formal definitions are given below.

$$\mathrm{MAE} = \frac{1}{N} \sum_{i=1}^{N} |y_i - \hat{y}_i| \tag{9}$$

$$\mathrm{RMSE} = \sqrt{\frac{1}{N} \sum_{i=1}^{N} (y_i - \hat{y}_i)^2} \tag{10}$$

$$\mathrm{MAPE} = \frac{100}{N} \sum_{i=1}^{N} \left| \frac{y_i - \hat{y}_i}{y_i} \right| \tag{11}$$

Here, $y_i$ denotes the ground-truth value, $\hat{y}_i$ is the predicted value, and $N$ is the number of samples.

## E  ADDITIONAL RESULTS FOR FORECASTING EVALUATION

In addition to the main experiments on traffic benchmarks, we also evaluate all methods on the WEATHER dataset to further examine their generalization under non-traffic domains. This dataset captures meteorological dynamics in Beijing, with two representative features: surface air pressure (*psur*) and relative humidity at $2\,\mathrm{m}$ (*rh2m*). The results reported in Table 5 show the forecasting performance of different approaches across short-, medium-, and long-horizon settings. Per-metric best values are boldfaced; lower is better.

As reported in Table 5, GAMAD also achieves state-of-the-art performance on the WEATHER dataset. For the feature *psur*, which follows highly regular seasonal cycles, most baselines already obtain low errors, yet GAMAD still reduces MAE by up to 13.2% compared with the strongest baseline (CGLM). While CGLM attains a lower RMSE on *psur*, GAMAD achieves lower MAE/MAPE; this is expected for near-stationary series with strong seasonality, where the $\ell_2$-based RMSE favors smoother predictors that suppress rare large errors, whereas our MAE-optimized,

Table 5: Multi-Step Forecasting Performance Comparison (Weather)

| | Hs | Metrics | HA | VAR | FC-LSTM | DCRNN | Graph WaveNet | AGCRN | GTS | MegaCRN | CGLM | **GAMAD** |
|---|---|---|---|---|---|---|---|---|---|---|---|---|
| WEATHER (psur) | H-3 | MAE | 28.52 | 1.56 | 2.06 | 1.06 | 1.00 | 1.03 | 1.04 | 1.11 | 0.94 | **0.92** |
| | | RMSE | 39.96 | 4.60 | 5.00 | 4.81 | 4.80 | 4.77 | 2.86 | 4.56 | 1.28 | **2.76** |
| | | MAPE | 3.00% | 0.16% | 0.21% | 0.11% | 0.10% | 0.11% | 0.11% | 0.11% | 0.10% | **0.09%** |
| | H-6 | MAE | 28.52 | 2.30 | 2.86 | 1.51 | 1.44 | 1.48 | 1.55 | 1.70 | 1.40 | **1.36** |
| | | RMSE | 39.96 | 5.36 | 5.80 | 5.23 | 5.13 | 5.11 | 3.39 | 5.32 | 1.75 | **3.18** |
| | | MAPE | 3.00% | 0.23% | 0.29% | 0.16% | 0.15% | 0.15% | 0.16% | 0.17% | 0.14% | **0.14%** |
| | H-12 | MAE | 28.52 | 3.09 | 3.71 | 2.37 | 2.25 | 2.31 | 2.49 | 2.66 | 2.27 | **2.17** |
| | | RMSE | 39.96 | 6.02 | 6.54 | 5.24 | 5.57 | 5.60 | 4.30 | 6.24 | 2.62 | **3.93** |
| | | MAPE | 3.00% | 0.31% | 0.38% | 0.24% | 0.23% | 0.24% | 0.25% | 0.27% | 0.23% | **0.22%** |
| WEATHER (q2m) | H-3 | MAE | 20.41 | 7.43 | 7.82 | 5.78 | 5.78 | 6.44 | 5.56 | 5.73 | 5.37 | **5.28** |
| | | RMSE | 23.65 | 10.24 | 11.13 | 8.62 | 8.67 | 9.47 | 8.10 | 8.52 | 6.69 | **6.59** |
| | | MAPE | 56.37% | 19.22% | 24.26% | 13.73% | 13.76% | 15.40% | 13.17% | 13.50% | 12.90% | **12.81%** |
| | H-6 | MAE | 20.41 | 12.09 | 12.69 | 8.27 | 8.21 | 9.57 | 8.03 | 8.21 | 7.69 | **7.46** |
| | | RMSE | 23.65 | 15.30 | 16.37 | 11.98 | 11.98 | 13.52 | 11.08 | 11.76 | 9.22 | **8.94** |
| | | MAPE | 56.37% | 33.41% | 42.60% | 20.06% | 19.92% | 24.11% | 19.56% | 20.17% | 19.10% | **18.69%** |
| | H-12 | MAE | 20.41 | 16.32 | 14.21 | 11.18 | 10.90 | 11.38 | 10.92 | 11.12 | 10.48 | **9.90** |
| | | RMSE | 23.65 | 19.76 | 18.01 | 15.75 | 15.30 | 15.79 | 14.48 | 15.48 | 12.11 | **11.45** |
| | | MAPE | 56.37% | 47.83% | 46.98% | 28.08% | 27.56% | 29.34% | 27.93% | 28.74% | 27.53% | **25.98%** |

drift-responsive design reduces typical errors. For the feature *rh2m*, where substantial short-term fluctuations are superimposed on seasonal trends, the task becomes considerably more challenging; here GAMAD shows even larger gains, reducing MAE by 10.5%, 9.2%, and 12.2% over CGLM on short-, medium-, and long-horizon predictions, respectively. These results highlight that while conventional GNNs fail to adapt to evolving dynamics and online baselines rely mainly on reactive updates, our drift-aware framework maintains robustness under both predictable and highly volatile temporal patterns.

# F  HYPERPARAMETER SENSITIVITY

We provide an extended study of GAMAD's hyperparameters in the online adaptation phase using the settings from the main text. Unless otherwise stated, we report MAE, RMSE, MAPE exhibit consistent trends.

**Learning rates $\alpha_{\text{ol}}$ and $\alpha_{\text{dl}}$.**  These subgraph updating rates have the largest impact (Fig. 5). We observe broad stability bands across the tested ranges, with degradation only at extreme values. A slightly larger $\alpha_{\text{dl}}$ (during detected drift) improves responsiveness to abrupt changes, whereas a more conservative $\alpha_{\text{ol}}$ (non-drift) avoids overfitting to noise.

**Epoch budgets $E_{\text{ol}}$ and $E_{\text{dl}}$.**  Varying the number of online updating epochs in non-drift and drift shows marginal effects: one to a few epochs already deliver most of the gains, and larger budgets yield diminishing returns without improving robustness.

**Drift-detection window size $W$.**  The detection window size governs how quickly the model identifies local distributional changes. Across the evaluated range, GAMAD remains stable, reflecting that even short windows (e.g., $W = 5$) effectively capture transient distribution shifts while providing natural resistance to noise.

**Maximum subgraph expansion size $N_{\text{sg}}^{\text{max}}$.**  The expansion size determines how far the dynamic subgraph grows around drifted nodes. Performance remains largely unchanged across different expansion limits, indicating that the correlation-guided hop expansion already focuses updates on the most relevant neighborhood. Although excessively large expansions may dilute the drift signal, the tested ranges show negligible sensitivity.

**Top-$K$ matching for pattern reuse.** For drift-pattern reuse, we match each newly detected subgraph against the top-$K$ most similar historical drift patterns. Since different choices of $K$ produce different numbers of matched patterns, we adopt a linearly decaying weighting scheme to ensure stable fusion: for a given $K$, the unnormalized weights are $[K, K-1, \ldots, 1]$, which are then normalized to sum to 1. This scheme assigns higher influence to more similar patterns while maintaining comparability across different $K$ values.

Empirically, the performance remains stable across the tested range of $K$, indicating that a small set of representative historical patterns is sufficient for reliable drift-guided adaptation. Increasing $K$ does not yield performance gains, as the decaying weights naturally suppress less relevant matches and prevent noisy historical patterns from dominating the adaptation process.

Across the tested ranges of $\alpha_{\mathrm{ol}}$, $\alpha_{\mathrm{dl}}$, epoch budgets, detection window sizes, expansion sizes, and top-$K$ matching choices, GAMAD maintains stable accuracy with only minor tuning. This low hyperparameter sensitivity aligns with our *graph-driven autonomous* design: node-level detection and correlation-conditioned subgraph updates localize learning to drift regions, while history-based reuse of patterns mitigates dependence on precise knob settings.

## G  NOISE ROBUSTNESS EVALUATION

We conduct an additional robustness study to evaluate the stability of GAMAD's online adaptation mechanism under noisy input conditions. To simulate real-world sensor disturbances, we inject controlled perturbations into the online-arriving slice $\mathcal{S}_t^{\mathrm{upd}}$ during testing.

**Noise model.** Each online slice is perturbed by a mixed noise process consisting of Gaussian fluctuations and sparse spike anomalies. Formally, the noisy slice $\widetilde{\mathcal{S}}_t^{\mathrm{upd}}$ is defined as:

$$\widetilde{\mathcal{S}}_t^{\mathrm{upd}} = \mathcal{S}_t^{\mathrm{upd}} + \epsilon_t^{\mathrm{Gauss}} + \epsilon_t^{\mathrm{Spike}}, \tag{12}$$

where

$$\epsilon_t^{\mathrm{Gauss}} \sim \mathcal{N}\Big(0,\ 0.1 \cdot \mathrm{std}(\mathcal{S}_t^{\mathrm{upd}})\Big), \tag{13}$$

$$\epsilon_t^{\mathrm{Spike}}(i) = \begin{cases} 0.5 \cdot \mathcal{S}_t^{\mathrm{upd}}(i), & \text{with probability } 0.02, \\ 0, & \text{otherwise.} \end{cases} \tag{14}$$

Gaussian noise models random sensor jitter, while spike perturbations simulate sporadic hardware or transmission faults. Perturbations are applied independently across time, modeling transient noise rather than persistent drift.

**Results.** Table 6 reports the overall online multi-step forecasting performance and drift-detection statistics on METR-LA. Here, MAE/MAPE/RMSE are averaged over the 12-step prediction horizons, consistent with the main evaluation protocol. Performance degradation remains within $\pm 0.5\%$ across all metrics, and the drift detection ratio remains stable, demonstrating that the window-based node-level detector effectively distinguishes transient noise from true drift events.

Table 6: Noise robustness of GAMAD under Gaussian + spike perturbations on METR-LA.

| Setting | MAE | MAPE | RMSE | Drift Steps | Non-Drift Steps |
|---|---|---|---|---|---|
| No Noise | 2.7057 | 0.0733 | 5.0847 | 4224 (61.77%) | 1811 (26.48%) |
| Gaussian + Spike | 2.7184 | 0.0731 | 5.0838 | 3474 (50.80%) | 2561 (37.45%) |
| $\Delta$ Change (%) | +0.47% | -0.27% | -0.02% | -17.78% | +41.44% |

**Summary.** Overall, the results demonstrate that GAMAD exhibits strong robustness under noisy conditions. Performance degradation remains below $0.5\%$ across all metrics, and the window-based node-level detector reduces false drift alarms by ignoring transient perturbations that do not form

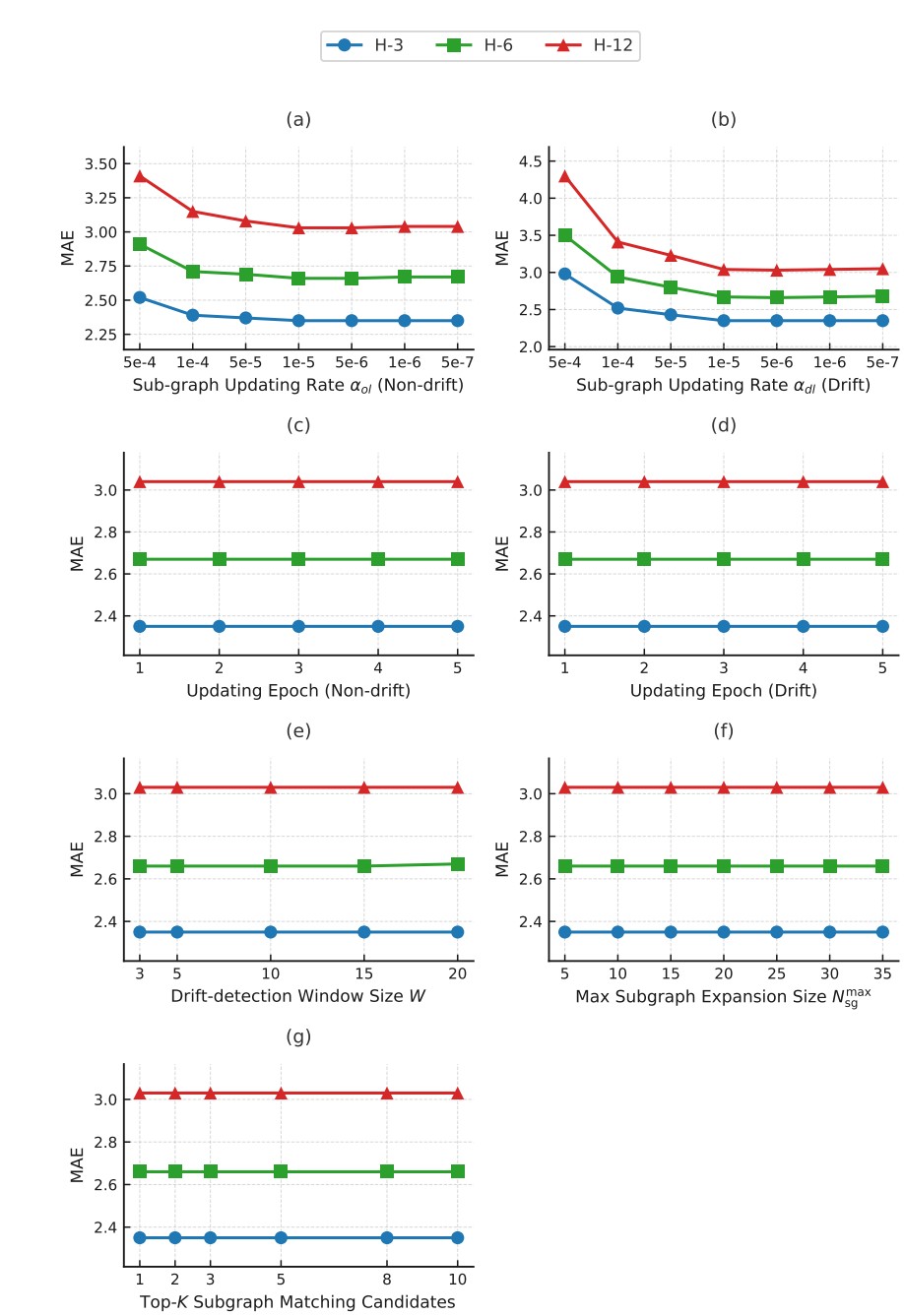

Figure 5: Sensitivity of GAMAD to key hyperparameters in online adaptation. Panels: (a) $\alpha_{\mathrm{ol}}$ (non-drift), (b) $\alpha_{\mathrm{dl}}$ (drift), (c) $E_{\mathrm{ol}}$ (non-drift), (d) $E_{\mathrm{dl}}$ (drift), (e) $W$, (f) $N_{\mathrm{sg}}^{\max}$, (g) top-$K$. Curves show large stable regions; learning rates dominate, while others have minor effects.

consistent distribution shifts. This confirms that the detector distinguishes noise from true drift, rather than being overly sensitive to random fluctuations.

Moreover, the robustness arises from both (1) the design of the graph-driven online adaptation module—including windowed drift detection, structure-aware subgraph expansion, and top-$K$ pattern reuse with linearly decaying weights—and (2) the strong generalization ability of the hierarchical

correlation graph learned offline. Together, these components ensure stable adaptation even when the online inputs are corrupted by Gaussian and sparse spike disturbances.

## H  STATISTICAL SIGNIFICANCE TESTING

We provide statistical significance tests to validate that the performance improvements of GAMAD over the strongest baseline (CGLM) are not due to random variation. For each metric (MAE, RMSE, MAPE) on METR-LA, we treat the three horizons (H-3, H-6, H-12) as paired samples and conduct paired two-tailed $t$-tests at the $0.05$ significance level. We report the mean differences (CGLM $-$ GAMAD), $t$-statistics and $p$-values in Table 7.

Table 7: Paired $t$-test results on METR-LA comparing CGLM and GAMAD.

| Metric | Mean diff. (CGLM$-$GAMAD) | $t$-statistic | $p$-value |
|--------|---------------------------|---------------|-----------|
| MAE    | 0.05                      | 8.66          | 0.013     |
| RMSE   | 0.11                      | 19.05         | 0.0027    |
| MAPE   | 0.157                     | 9.40          | 0.011     |

All three metrics yield $p$-values below $0.05$, indicating that the improvements of GAMAD over CGLM on METR-LA are statistically significant rather than due to random fluctuations.

## I  RUNTIME AND COMPLEXITY ANALYSIS

We report both empirical runtime measurements and a concise complexity analysis, complementing the discussions in the main text. All experiments are conducted on a single NVIDIA A100 GPU.

### EMPIRICAL RUNTIME

Table 8 summarizes the offline training time per epoch and the online adaptation latency per update step. Despite introducing hierarchical graph construction and dynamic updates, our method maintains comparable offline runtime to other graph-based baselines. During online adaptation, the average update latency of $1.27$ s is *significantly smaller* than the updating interval of the METR-LA dataset (5 minutes), and thus comfortably satisfies real-time constraints for streaming prediction.

Table 8: Empirical runtime comparison (seconds).

| Method | Offline (s/epoch) | Online latency (s/step) |
|--------|-------------------|-------------------------|
| GTS (no predefined graph) | 49.26 | – |
| CGLM (dynamic graph) | 55.74 | 1.11 |
| GAMAD(ours) | 64.63 | **1.27** |

### COMPLEXITY ANALYSIS

**Global correlation graph construction.**  The global hierarchical correlation matrix is constructed only once during training with complexity $\mathcal{O}(N^2)$, where $N$ is the number of data streams.

**Online drift adaptation.**  Each adaptation operates on a dynamically expanded local subgraph of size $k$ around detected drift nodes. The update cost is therefore $\mathcal{O}(k)$ rather than $\mathcal{O}(N)$, and in practice $k < 20$.

**Dynamic graph updates.**  Since online updates modify only local correlations, the method avoids re-computing global adjacency matrices and scales well to larger graphs. Both METR-LA (207 nodes) and PEMS-BAY (325 nodes) exhibit stable runtime under our online learning pipeline.

