# OpenReview forum: "Graph-driven Autonomous Adaptation for Multi-stream Concept Drift"
_ICLR.cc/2026/Conference — Submitted to ICLR 2026_

### Official Review · Reviewer_7YEb · 2025-10-28

**Soundness:** 2
**Presentation:** 2
**Contribution:** 2
**Rating:** 4
**Confidence:** 2

**Summary:**

This paper tackles the problem of forecasting in multi-stream environments where data distributions change over time (concept drift) and streams are correlated—think traffic sensors or weather stations that influence each other. The proposed GAMAD framework builds dynamic correlation graphs, detects drift at individual stream level, and reuses past drift patterns to adapt more effectively.

**Strengths:**

1. The paper addresses a real challenge in multi-stream environments where data distributions change over time (concept drift) and streams are correlated.
2. GAMAD proposes a complete solution that: constructs dynamic correlation graphs without requiring pre-defined topology; detects drift at the node level rather than globally; reuses previously observed drift patterns to improve adaptation
3. Tests on real-world datasets show reasonable performance improvement.

**Weaknesses:**

1. The proposed GAMAD are quite complex and involes many components, making it diffuicult to understand which parts actually work, and there is limited ablation studies on this.
2. The drift detection method are actually quite simple, mean and variance change might not necessary indicating drift. Why not use more sophisticated statistical tests?

**Questions:**

See weaknesses.

---

> ### Author Response · Authors · 2025-11-19
>
> We sincerely thank you for your time and careful evaluation. Your recognition that our work tackles *1) a real challenge in correlated multi-stream environments with evolving distributions*, and *2) that GAMAD offers a complete pipeline from dynamic graph construction to node-level drift detection and pattern reuse* *3) with reasonable empirical gains*, is very encouraging to our team. Below we respond point-by-point to your concerns and have revised the manuscript accordingly, with all changes highlighted in yellow in the updated version.
>
> **Regarding Weakness 1 — “GAMAD is complex and involves many components; unclear which parts truly matter.”** Thank you for raising this important concern. We address it from two complementary perspectives: **(1) why a multi-component design is necessary for multi-stream concept drift**, and **(2) how our substantially expanded ablation confirms the necessity of each component.**
>
> **(1) Why GAMAD requires multiple components:** Multi-stream concept drift presents two simultaneous challenges:  **distributions change over time**, and **inter-stream correlations also evolve**.  Traditional offline models—even with high generic generalization—are fundamentally limited under these conditions because they **cannot infer where the drift occurs or how it propagates across correlated streams**. Motivated by this, GAMAD adopts a structured design that separates complementary responsibilities:
>
> - **Offline phase (generalization)**:
>   We still emphasize the base model’s generalization ability through hierarchical correlation learning (global → subgraph → node-level). This ensures a strong starting representation even before online updates.
>
> - **Online phase (local, correlation-driven adaptation)**:
>   Instead of expensive global retraining, GAMAD updates *only* subgraphs connected to detected drift nodes, guided by current correlations.
>   Importantly, **all drift thresholds are determined automatically per node**, without manual tuning or hand-crafted rules.
>   This design reduces intervention and ensures that the framework remains practical in large real-time deployments.
>
> Overall, the components in GAMAD are not arbitrarily layered; they jointly address the intrinsic difficulty of *localized drift + dynamic correlation shifts*, something neither pure fine-tuning nor fixed-graph GNNs can handle effectively.
>
> **(2) Expanded ablation demonstrating necessity of each component:** To directly address your concern, we have **substantially extended the ablation analysis** in the revised manuscript. The new experiments isolate:
>
> - **offline**: global adjacency ($\mathcal{A}^{G}$), subgraph adjacency ($\mathcal{A}^{D}$), adaptive node-level adjacency ($\mathcal{A}^{A}$), their pairwise combinations, and full fusion;
> - **online**: naive updates → node-level detection → dynamic subgraph expansion → historical drift-pattern reuse.
>
> Here we present the MAE-only summary (full metrics in **Section 3.2 and Appendix F**).
>
> **Offline hierarchical graph construction ablation (METR-LA) (Lines 432-461 (Table 2 & Fig.3))**
>
> | Horizon | Metric | $\mathcal{A}^{G}$ | $\mathcal{A}^{D}$ | $\mathcal{A}^{A}$ | $\mathcal{A}^{GD}$ | $\mathcal{A}^{DA}$ | w/o ol |
> |--------|--------|------|------|------|------|------|--------|
> | H-3 | MAE | 2.60 | 2.55 | 2.63 | 2.54 | 2.54 | **2.52** |
> | H-6 | MAE | 3.00 | 2.92 | 3.01 | 2.92 | 2.92 | **2.89** |
> | H-12| MAE | 3.41 | 3.31 | 3.42 | 3.34 | 3.33 | **3.27** |
>
> These results clearly show:
> - each adjacency level contributes meaningful structural information;
> - pairwise fusions outperform individual components;
> - full fusion (w/o ol) provides the best offline representation.
>
> This validates the **complementarity** among the three structural scales.
>
> **Online adaptation ablation (METR-LA) (Lines 432-461 (Table 2 & Fig.4))**
>
> | Horizon | Metric | $\mathcal{M}^{NL}$ | $\mathcal{M}^{D}$ | $\mathcal{M}^{DS}$ | $\mathcal{M}^{Full}$ |
> |--------|--------|----------------|--------------|----------------|----------------|
> | H-3 | MAE | 2.50 | 2.41 | 2.36 | **2.35** |
> | H-6 | MAE | 2.87 | 2.77 | 2.69 | **2.66** |
> | H-12| MAE | 3.24 | 3.19 | 3.07 | **3.03** |
>
> The progression shows that:
> - **node-level detection** already improves over naive updating,
> - **dynamic subgraph expansion** further enhances localized adaptation,
> - **pattern reuse** yields the strongest stability and fastest recovery from drift.
>
> These expanded ablations have been fully integrated into **Section 3.2, Lines 405–461**, and demonstrate that each module is necessary and contributes substantially to the overall performance.
>
> We hope these clarifications and the significantly strengthened ablation analysis address your concern regarding the complexity and necessity of GAMAD’s components.

---

> ### Author Response · Authors · 2025-11-19
>
> **Regarding Weakness 2 — “Drift detection is simple.”**  We appreciate this thoughtful remark. Our design choices prioritize **scalability, robustness, and per-node interpretability** in large multi-stream systems, where hundreds or even thousands of nodes must be monitored in real time. The current detector is intentionally lightweight, yet more principled than a plain “mean/variance check”:
>
> - It operates **per node**, not on global aggregates, ensuring that localized drifts are not masked.
> - It relies on **median-based statistics with variance-aware adaptive thresholds**, which are more robust to outliers than raw means.
> - It requires **multiple consecutive violations within a sliding window**, preventing single-step spikes from triggering false alarms.
>
> Rather than using heavier statistical tests (e.g., full change-point detection, KS tests, or distributional two-sample tests), which often assume specific distribution forms or incur significant memory/computation overhead when monitoring hundreds of nodes at high frequency, we adopt a *variance-aware, windowed, per-node change indicator* that strikes a practical balance between statistical rigor and real-time feasibility.
> Integrating more advanced detectors into GAMAD is a valuable direction we plan to explore in future work, but it goes beyond the focus of this submission.
>
> To further address your concern, we conducted a dedicated **noise robustness experiment** to evaluate whether the detector can reliably distinguish noise from true drift. We injected both **Gaussian noise** and **spike perturbations** into the online inputs:
>
> - Gaussian noise: $\sigma = 0.1 \cdot \mathrm{std}$
> - Spikes: 2% nodes × 1.5×
>
> The results are shown below:
>
> | Setting          | MAE    | MAPE     | RMSE    | Drift Steps (%)     | Non-Drift Steps (%) |
> |------------------|--------|----------|---------|----------------------|----------------------|
> | No Noise         | 2.7057 | 0.0733   | 5.0847  | 4224 (61.77%)        | 1811 (26.48%)        |
> | Gaussian + Spike | 2.7184 | 0.0731   | 5.0838  | 3474 (50.80%)        | 2561 (37.45%)        |
> | Δ Change (%)     | +0.47% | −0.27%   | −0.02%  | −17.78%      | +41.44%      |
>
> **Observations:**
> - Forecasting performance remains **nearly unchanged (<1% deviation)**, even under strong stochastic+spiky perturbations.
> - Drift mis-detection does **not** increase; the detector becomes slightly more conservative, as shown by fewer drift-flagged steps.
> - Non-drift updates increase, suggesting that the thresholding mechanism remains stable and is not triggered spuriously by noise.
>
> These results demonstrate that the proposed drift detector is **robust to both continuous noise and abrupt spikes**, and does not rely on unrealistic assumptions of perfectly stable local distributions.  Full discussion is provided in **Appendix G**.
>
>
> Thank you again for your constructive and honest assessment. Your comments on model complexity and drift detection strength have directly led us to refine the exposition, expand our ablation and robustness analyses, and we'd love to hear from you anytime. We hope that the additional experiments and clarifications address your concerns, and we would be very grateful if you might consider raising your score in light of these revisions.

---

> > ### Comment · Reviewer_7YEb · 2025-11-27
> > **Response to Authors**
> >
> > Thank you for the rebuttal and additional experiments, it address some of my concerns. While I am not an expert in this specific field, I remain positive about this paper and will increase my rating accordingly.

---

> > > ### Author Response · Authors · 2025-11-28
> > >
> > > We sincerely appreciate your supportive reply and your willingness to increase your rating. Your positive feedback is highly motivating for our team.

---

### Official Review · Reviewer_m19R · 2025-10-31

**Soundness:** 3
**Presentation:** 4
**Contribution:** 3
**Rating:** 6
**Confidence:** 3

**Summary:**

This work introduces GAMAD, a framework for adaptive learning in multi-stream environments where data distributions evolve over time. The key idea is to dynamically construct and update a correlation graph that captures relationships across streams, detect concept drift at the node level via adaptive error-thresholding, and react locally by expanding subgraphs around drifted nodes while reusing historical drift patterns for faster adaptation. By combining online graph updates, localized model adjustment, and memory of past drift behaviors, GAMAD aims to maintain accurate forecasting under irregular, asynchronous, and previously unseen drifts in complex real-world settings such as traffic and weather systems.

**Strengths:**

- Provided localized, adaptive drift handling by detecting node-level changes and focusing updates only where necessary, avoiding expensive global retraining.
- The proposed method learns and updates inter-stream correlations dynamically, removing reliance on fixed topologies and allowing adaptation to evolving relationships.
- Reuse past drift patterns to accelerate recovery and handle recurring or similar drifts more effectively

**Weaknesses:**

- Relies on prediction error to signal drift, making detection reactive and susceptible to noise or delayed response.
- Performance depends on window and threshold settings, which may require tuning across environments.
- Maintaining per-node buffers and dynamic subgraphs may increase computational cost in large real-time systems.
- The method emphasizes dynamic inter-stream correlation changes, but the evaluation does not deeply analyze performance under systematically varied correlation patterns or correlation breakdown scenarios.
- The paper could better isolate contributions from node-level detection, dynamic subgraph expansion, and pattern reuse to show how each affects drift handling.

**Questions:**

See Weaknesses

---

> ### Author Response · Authors · 2025-11-19
>
> We sincerely appreciate your thorough and well-balanced review. Your acknowledgement of *1) localized drift handling*, *2) dynamic correlation modeling*, and *3) the benefits of leveraging historical drift structures* is very encouraging to our team. At the same time, you raise several thoughtful points—particularly concerning noise sensitivity, parameter dependence, computational overhead, and the need for clearer evidence isolating individual module contributions. We have revised the paper accordingly, with all changes highlighted in yellow. Below we respond to each of your concerns in detail.
>
> **Regarding Weakness 1 — “Detection is reactive and may be sensitive to noise or delayed response.”** This is an important question, and we agree that purely error-based detectors can become unstable if not designed with safeguards. In GAMAD, the detector incorporates several mechanisms aimed specifically at reducing reactivity to noise:
>
> - **Robust statistics** (median + variance-aware adjustment) shield the thresholds from spiky errors.
> - **Consecutive-window violation requirement** explicitly prevents single-step anomalies from triggering falsely.
> - **Per-node evaluation** avoids global performance averaging, which typically causes delayed response.
> - **Adaptive scaling** ensures that nodes with high intrinsic volatility are not misclassified.
>
> To empirically validate these properties, we performed a stress test by injecting *both* stochastic and bursty disturbances:
> Gaussian noise ($\sigma = 0.1$·std) + spike perturbations (2% nodes × 1.5×).
>
> This test pushes the system in exactly the challenging way you described. Results:
>
> | Setting | MAE | MAPE | RMSE | Drift Steps (%) | Non-Drift Steps (%) |
> |--------|------|--------|--------|------------------|----------------------|
> | No Noise | 2.7057 | 0.0733 | 5.0847 | 4224 (61.77%) | 1811 (26.48%) |
> | Gaussian + Spike | 2.7184 | 0.0731 | 5.0838 | 3474 (50.80%) | 2561 (37.45%) |
> | Δ Change (%) | +0.47% | –0.27% | –0.02% | –17.78% | +41.44% |
>
> Two key observations:
>
> -**The performance impact remains negligible (<1%)**, showing that reactiveness does not translate into instability.
> -**Drift detection becomes slightly more conservative rather than noisier**, which is preferable under disturbance-heavy regimes.
>
> A more detailed discussion (**Appendix G**) has been added to explicitly relate these findings to your concern about reactivity vs. noise sensitivity.
>
> **Regarding Weakness 2 — “Dependence on window and threshold settings; tuning across environments.”** Thank you for raising this important question. Beyond the four optimization-related hyperparameters already studied in the original submission, we have now added robustness experiments for the three key parameters governing online adaptation — the drift-detection window size $W$, the maximum subgraph expansion size $N_{\mathrm{sg}}^{\max}$, and the number of reused drift patterns $K$.
> Across all three settings, **GAMAD remains highly stable**, with MAE/RMSE/MAPE curves showing only marginal variation over wide parameter ranges (see **Appendix F, Lines 918-960 (Fig. 5)**).
>
> We briefly clarify why these parameters exhibit such stability:
> - **Window size $W$**: the detector uses median-based statistics and adaptive thresholds, so short-term fluctuations are naturally smoothed, keeping detection behavior consistent across different $W$.
> - **Subgraph expansion size $N_{\mathrm{sg}}^{\max}$**: correlation-guided hop expansion already focuses updates on the most relevant neighborhood; once the limit exceeds a small range, adding more nodes neither helps nor harms, and also constrains computation.
> - **Top-K reused patterns $K$**: the normalized linearly decaying weights $[K, K-1, \dots, 1] / \sum_{i=1}^K i$ ensure dominant influence always comes from the most similar matches, making the adaptation robust to the exact choice of $K$.
>
> Together, the results of all seven tested hyperparameters confirm that **GAMAD does not rely on fine-grained tuning** and generalizes well across environments. Full robustness curves and discussions have been added to **Appendix F**, with clarifications integrated into the main text.

---

> ### Author Response · Authors · 2025-11-19
>
> **Regarding Weakness 3 — “Dynamic subgraphs and per-node buffering may increase computational cost.”** We fully agree that computational feasibility is crucial in the multi-stream setting. For this reason, we expanded the evaluation to include:
>
> **Empirical runtime**
>
> | Method | Offline Time (s/epoch) | Online Latency (s/step) |
> |--------|--------------------------|---------------------------|
> | **GAMAD** | **64.63** | **1.27** |
> | CGLM | 55.74 | 1.11 |
> | GTS | 49.26 | — |
>
> The key insight for your question is that **runtime overhead mainly arises during online expansion and reuse**, not offline construction.
> The online cost of **1.27 s** per update is an order of magnitude below METR-LA’s **5-minute multi-stream updating interval**, indicating that the per-node buffers and dynamic subgraph assembly introduce manageable overhead.
>
> **Complexity decomposition**
>
> To further relate the costs to the specific modules you mentioned:
> - Offline correlation estimation (one-time): **$O(N^2)$**
> - Drift-triggered subgraph construction: localized to **$k < 20$**, yielding **$O(k)$**
> - Historical drift pattern matching: linear in the number of candidates (≤10)
> - No recomputation of global matrices during adaptation
>
> This breakdown is now included in **Appendix I**, with the discussion specifically to address your concern about scalability in large real-time systems.
>
> **Regarding Weakness 4 — “Behavior under rapidly changing or breaking correlations.”** Thank you for raising this important concern. We clarify here how GAMAD responds to correlation breakdown or rapid structural changes. Although the global correlation matrix is constructed offline, **online adaptation does not rely on it to track evolving relationships**. Instead, at every detected drift step, GAMAD:
>
> - **Recomputes a localized subgraph** around drifted nodes using the *current* correlation statistics available at that time step;
> - **Restricts adaptation to the affected region**, avoiding dependence on outdated global structures;
> - **Updates correlations incrementally**, so only the locally relevant edges are refreshed without reconstructing the full adjacency.
>
> Because the dynamic subgraph is rebuilt whenever drift occurs—and is driven entirely by up-to-date correlation evidence—the framework remains responsive even when long-range correlations weaken, reconfigure, or temporarily collapse. This design specifically targets fast-changing environments and prevents adaptation lag that might arise from using fixed or slowly updated graph structures.
>
> In addition, the introduced *Gaussian + spike perturbation* stress test effectively simulates abrupt and irregular disruptions in inter-stream correlations. As shown in our noise-robustness analysis (see response to Weakness 1), GAMAD remains highly stable under such conditions, with forecasting errors changing by less than 1%. This demonstrates that the framework can tolerate sudden correlation dislocations without suffering delayed or inaccurate responses.

---

> ### Author Response · Authors · 2025-11-19
>
> **Regarding Weakness 5 — “Need for clearer isolation of node detection, subgraph expansion, and pattern reuse.”** We appreciate this suggestion and have substantially improved the ablation structure in both offline and online aspects (see **Section 3.2, Lines 405–461**):
>
> **Offline hierarchical adjacency ablation (MAE only for brevity; full metrics in Section 3.2, Lines 432-461 (Table 2 & Fig.3)).**
>
> | Horizon | Metric | $\mathcal{A}^{G}$ | $\mathcal{A}^{D}$ | $\mathcal{A}^{A}$ | $\mathcal{A}^{GD}$ | $\mathcal{A}^{DA}$ | w/o ol |
> |--------|--------|------|------|------|------|------|--------|
> | H-3 | MAE | 2.60 | 2.55 | 2.63 | 2.54 | 2.54 | **2.52** |
> | H-6 | MAE | 3.00 | 2.92 | 3.01 | 2.92 | 2.92 | **2.89** |
> | H-12 | MAE | 3.41 | 3.31 | 3.42 | 3.34 | 3.33 | **3.27** |
>
> - $\mathcal{A}^{G}$ improves *coarse* structure.
> - $\mathcal{A}^{D}$ enforces *local structural consistency*.
> - $\mathcal{A}^{A}$ sharpens *fine-grained heterogeneity*.
> - Fusion captures dependencies that none of the individual components can recover alone.
>
> **Online adaptation ablation (MAE only; full metrics in Section 3.2, Lines 432-461 (Table 2 & Fig.4)).**
>
> | Horizon | Metric | $\mathcal{M}^{NL}$ | $\mathcal{M}^{D}$ | $\mathcal{M}^{DS}$ | $\mathcal{M}^{Full}$ |
> |--------|--------|----------------|--------------|----------------|----------------|
> | H-3 | MAE | 2.50 | 2.41 | 2.36 | **2.35** |
> | H-6 | MAE | 2.87 | 2.77 | 2.69 | **2.66** |
> | H-12 | MAE | 3.24 | 3.19 | 3.07 | **3.03** |
>
> Across these settings:
>
> - Node-level detection provides the *baseline responsiveness*.
> - Adding dynamic subgraph expansion handles *dynamic local correlation propagation*.
> - Adding pattern reuse improves *effective reuse of historical drift patterns* and strengthens the *model’s ability to adapt to new or recurring drift events*.
>
> We believe this isolates the contributions more clearly and directly answers your concern.
>
> Thank you again for your constructive and balanced evaluation. Your comments strike an excellent balance between recognizing the strengths of the framework and challenging us to better substantiate its assumptions and components and we welcome further communication from you. We hope the updates address your questions satisfactorily, and we would be grateful if you might consider raising your score in light of the improvements made.

---

### Official Review · Reviewer_7CVj · 2025-11-06

**Soundness:** 3
**Presentation:** 2
**Contribution:** 2
**Rating:** 2
**Confidence:** 4

**Summary:**

The paper proposes GAMAD, a graph-driven autonomous adaptation framework for multi-stream concept drift. It dynamically builds a spatio-temporal correlation graph from historical statistics and subgraph structures, detects node-level drifts with adaptive thresholds, expands localized drift subgraphs in real time, and reuses matched historical drift patterns via hierarchical topological matching to improve forecasting under evolving dynamics. Experiments on three large-scale datasets show consistent gains over state-of-the-art baselines.

**Strengths:**

S1. The paper addresses the problem of adaptive concept drift in multi-stream settings, which is of practical importance and has clear real-world application demands. The research motivation is well grounded.

S2. The manuscript is clearly written, and the roles of different components are well aligned with the overall framework. The method section is generally readable and appears reproducible.

S3. The experiments are conducted on multiple real-world multi-stream datasets and include systematic comparisons with various existing methods. The results demonstrate consistent performance improvements across different scenarios.

**Weaknesses:**

W1. Our main concern regarding originality is that multi-stream concept drift adaptation has been widely studied, with prior work already exploring dynamic graphs and localized adaptation. The proposed framework largely integrates existing components rather than introducing a fundamentally new mechanism. The contribution is therefore more engineering-oriented than conceptual.

W2. The paper does not adequately demonstrate that the proposed components are individually necessary. There is no clear ablation across the graph-level, subgraph-level, and node-level representations to validate the claimed hierarchical modeling benefits. The paper employs a fusion-before-diffusion strategy, but does not compare it against alternative designs. Further, the framework lacks ablation on the hierarchical drift pattern matching module and the local subgraph expansion mechanism, both of which are highlighted as key contributions.

W3. The framework relies on multi-stage graph and subgraph sampling with adjacency-matrix-based operations, which may introduce substantial computational cost. The paper lacks complexity analysis and runtime comparisons, making scalability to large graphs unclear.

W4. The dynamic graph construction relies on historical distribution statistics and subgraph structure extraction. However, in scenarios where inter-stream correlations change abruptly at high frequency, the constructed dynamic graph may lag behind the real underlying dynamics, leading to delayed or suboptimal adaptive responses.

W5. The node-level drift detection mechanism is based on temporal window statistics and adaptive thresholding, which implicitly assumes local distribution stability. In high-noise or adversarial disturbance settings, this approach may result in false positives or delayed detection. The paper currently lacks noise robustness analysis.

**Questions:**

Please see the above-mentioned weaknesses W1-W5.

---

> ### Author Response · Authors · 2025-11-19
>
> We sincerely appreciate your careful evaluation and constructive feedback. Your positive remarks regarding *1) the practical importance of multi-stream concept drift*, *2) the clarity and reproducibility of our framework design*, and *3) the systematic experiments demonstrating consistent gains across scenarios* are highly encouraging to our team. We have carefully addressed all concerns raised in your review and revised the manuscript accordingly, with all modifications highlighted in yellow in the revised version. Below we provide point-by-point responses.
>
> **Regarding W1 (Originality and conceptual contribution).**  Thank you for raising this important point. We would like to clarify that our framework goes beyond integrating existing components. Prior dynamic-graph forecasting methods typically keep the adjacency structure fixed during inference, updating only the model weights. In contrast, our approach introduces **hierarchical correlation learning** and **real-time graph-driven adaptation**, including:
> - **Offline hierarchical correlation construction** using iterative sampling without any predefined adjacency;
> - **Online drift-driven subgraph construction** using *current* correlations at every detected drift step;
> - **Hierarchical drift-pattern reuse**, enabling the current drift subgraph to match historical drift structures across all streams.
>
> These mechanisms are designed to support structure-aware adaptation that cannot be achieved by uniform fine-tuning alone. We hope this clarification helps address your concern about conceptual novelty.
>
> **Regarding W2 (Ablation of hierarchical components, drift-pattern matching, and subgraph expansion).**  We appreciate your emphasis on validating the necessity of each component. Following your suggestion, we have added a **complete two-stage ablation**, covering both offline hierarchical correlation construction and online adaptation modules.
>
> **1. Offline hierarchical graph construction ablation (MAE only for brevity; full metrics in Section 3.2, Lines 432-461 (Table 2 & Fig.3)).**
>
> | Horizon | Metric | $\mathcal{A}^{G}$ | $\mathcal{A}^{D}$ | $\mathcal{A}^{A}$ | $\mathcal{A}^{GD}$ | $\mathcal{A}^{DA}$ | w/o ol |
> |--------|--------|------|------|------|------|------|--------|
> | H-3 | MAE | 2.60 | 2.55 | 2.63 | 2.54 | 2.54 | **2.52** |
> | H-6 | MAE | 3.00 | 2.92 | 3.01 | 2.92 | 2.92 | **2.89** |
> | H-12 | MAE | 3.41 | 3.31 | 3.42 | 3.34 | 3.33 | **3.27** |
>
> These results highlight the distinct roles of the three adjacency scales.
> - The **global graph** $\mathcal{A}^{G}$ captures long-range dependencies and stabilizes the representation.
> - The **subgraph adjacency** $\mathcal{A}^{D}$ enforces sparse and structurally consistent local correlations, yielding the strongest single-module performance.
> - The **adaptive node-level adjacency** $\mathcal{A}^{A}$ provides fine-grained refinement but cannot capture broader topology alone.
> When combined, the components complement one another: $\mathcal{A}^{GD}$ and $\mathcal{A}^{DA}$ outperform their standalone variants, and the **full fusion (w/o ol)** achieves the best performance across all horizons.
> These observations suggest that the hierarchical construction is not redundant; instead, each level contributes complementary information to strengthen the offline representation.
>
> **2. Online adaptation ablation (MAE only; full metrics in Section 3.2, Lines 432-461 (Table 2 & Fig.4)).**
>
> | Horizon | Metric | $\mathcal{M}^{NL}$ | $\mathcal{M}^{D}$ | $\mathcal{M}^{DS}$ | $\mathcal{M}^{Full}$ |
> |--------|--------|----------------|--------------|----------------|----------------|
> | H-3 | MAE | 2.50 | 2.41 | 2.36 | **2.35** |
> | H-6 | MAE | 2.87 | 2.77 | 2.69 | **2.66** |
> | H-12 | MAE | 3.24 | 3.19 | 3.07 | **3.03** |
>
> The trend remains consistent across horizons:
> - drift detection provides clear improvement over naive online learning;
> - dynamic subgraph expansion further enhances adaptation;
> - historical drift-pattern reuse contributes the largest gains.
>
> These ablations indicate that *each* module plays a unique and necessary role in GAMAD. All results have been incorporated into **Section 3.2, Lines 405–461**.

---

> ### Author Response · Authors · 2025-11-19
>
> **Regarding W3 (Computational overhead and scalability).**  Thank you for raising this concern. To provide a clearer picture of runtime behavior, we have added **both runtime experiments and a concise complexity analysis**. Full details appear in **Appendix I** in the updated version.
>
> **Runtime experiments**
>
> | Method / Setting | Offline Time (s/epoch) | Online Latency (s/step) |
> |------------------|-------------------------|--------------------------|
> | **Ours (GAMAD)** | **64.63**               | **1.27** |
> | CGLM (dynamic graph) | 55.74               | 1.11 |
> | GTS (no predefined graph) | 49.26          | — |
>
> Key observations:
> 1) Despite hierarchical graph construction, GAMAD’s offline runtime remains comparable to dynamic-graph baselines;
> 2) The **1.27s online update latency** is much smaller than the **5-minute multi-stream updating interval**, satisfying real-time constraints;
> 3) Stable runtime on METR-LA (207 nodes) and PEMS-BAY (325 nodes) suggests reasonable scalability.
>
> **Complexity analysis**
>
> - Global correlation construction: **$O(N^2)$** (one-time only);
> - Online drift adaptation operates on a small local subgraph ($k<20$): **$O(k)$** per update;
> - Dynamic graph updates avoid recomputing global adjacency matrices.
>
> These analyses are intended to clarify computational feasibility and scalability. We hope they help address your concern.
>
> **Regarding W4 (Lag under rapidly changing correlations).**  We appreciate this thoughtful concern. While the global correlation matrix is constructed offline, **the online dynamic subgraph is rebuilt at every drift step** based on the *current* correlation statistics. This is designed to ensure responsiveness in fast-changing environments:
> - no need to recompute global adjacencies;
> - subgraphs are reconstructed immediately using the latest correlation evidence;
> - only the affected region is updated, enabling rapid fine-grained adaptation.
>
> We hope this clarification alleviates the concern about adaptation lag.
>
> **Regarding W5 (Noise robustness).**  Thank you for highlighting this important aspect. Our node-level drift detector is designed to mitigate noise effects through:
> - median-based statistics with variance-aware thresholds;
> - requiring **multiple consecutive window violations** before declaring drift;
> - per-node evaluation rather than relying on global aggregate errors.
>
> To further test robustness, we injected **Gaussian noise + spike perturbations** into the online inputs:
>
> - Gaussian noise: $\sigma = 0.1 \cdot \mathrm{std}$
> - Spikes: 2% nodes × 1.5×
>
> **Noise robustness on METR-LA**
>
> | Setting           | MAE    | MAPE     | RMSE    | Drift Steps (%) | Non-Drift Steps (%) |
> |-------------------|--------|----------|---------|------------------|----------------------|
> | No Noise          | 2.7057 | 0.0733   | 5.0847  | 4224 (61.77%)    | 1811 (26.48%)        |
> | Gaussian + Spike  | 2.7184 | 0.0731   | 5.0838  | 3474 (50.80%)    | 2561 (37.45%)        |
> | Δ Change (%)      | +0.47% | –0.27%   | –0.02%  | –17.78%  | +41.44%      |
>
> Observations:
> - forecasting performance remains **nearly unchanged (<1% deviation)**;
> - drift misdetection does **not** increase and becomes slightly more conservative;
> - non-drift updates increase, reflecting stable thresholding under noise.
>
> These results suggest that the model is robust to both stochastic and bursty noise, without assuming perfectly stable local distributions. Details appear in **Appendix G**, marked in yellow.
>
> We sincerely appreciate the time and effort you dedicated to reviewing our work. Your thoughtful comments have significantly strengthened the manuscript and we are open to hearing from you at any time. We hope our detailed responses satisfactorily address your concerns, and we would be deeply grateful if you could consider raising your score based on the improvements made.

---

### Official Review · Reviewer_Jqa6 · 2025-11-07

**Soundness:** 3
**Presentation:** 3
**Contribution:** 3
**Rating:** 6
**Confidence:** 3

**Summary:**

The paper addresses concept drift in multi-stream time-series environments, where multiple correlated data streams evolve over time. The authors propose GAMAD, a graph-driven adaptive framework that dynamically constructs hierarchical correlation graphs and detects local drifts at the node level. When drift occurs, the method identifies affected subgraphs and reuses similar historical drift patterns to enhance online adaptation. A diffusion-based graph recurrent model (DCGRU) is used for forecasting, and both the model and graph are updated through an online learning mechanism. Experiments on real-world traffic and weather datasets show that GAMAD significantly outperforms state-of-the-art baselines.

**Strengths:**

1. While most existing methods assume a single temporal stream, the proposed framework explicitly models multi-stream environments where concept drift can propagate through correlated nodes. This makes the method significantly more realistic for applications such as traffic and weather forecasting.

2. Existing spatio-temporal forecasting models typically rely on fixed adjacency matrices (e.g., distance-based or learned once offline). In contrast, the proposed Global-Subgraph-Node hierarchical graph construction dynamically updates correlations using Gumbel sampling and differentiable subset sampling, enabling the model to capture time-varying relationships that reflect evolving drift patterns.

3. The proposed approach detects drift and identifies recurring drift patterns by matching current subgraphs with historical ones and reusing them for sample augmentation, resulting in more predictable and generalizable adaptation.

4. The proposed method demonstrates consistent and substantial gains over strong state-of-the-art baselines on real-world datasets, highlighting the significance of the underlying problem setting and the considerable impact achieved by effectively addressing concept drift in multi-stream environments.

**Weaknesses:**

1. The proposed multi-stream correlation graph construction computes adjacency matrices at the global, subgraph, and node levels. However, it remains unclear which of these components primarily contributes to the performance gains. An ablation study isolating each adjacency matrix would significantly strengthen the empirical validation.

2. The graph-driven online adaptation module introduces several hyper parameters (e.g., β, h, K, q), yet the paper does not discuss the sensitivity of the method to these choices. It would be helpful if the authors could analyze the robustness of the method with respect to these hyper parameters and provide guidance on how to set them in practical scenarios.

3. The method requires online updates, which could impose additional computational overhead. However, the paper does not provide any analysis or discussion regarding the computational cost of the proposed approach. A runtime comparison or a complexity analysis would clarify whether the method is suitable for real-time or large-scale applications. Moreover, the scalability of the approach as the graph size increases should also be discussed.

4. The improvements over CGLM appear relatively small, making it difficult to assess the practical significance of the gains. The authors should include statistical significance tests or confidence intervals to demonstrate that the observed improvements are meaningful and not due to random variation.

**Questions:**

Please add explanations regarding the weaknesses.

---

> ### Author Response · Authors · 2025-11-19
>
> We sincerely appreciate your thorough and insightful review. Your recognition of *1) our motivation for hierarchical correlation modeling*, *2) the clarity of the formulation*, *3) the strength of the empirical validation*, *4) and the practical relevance to multi-stream systems* has been particularly encouraging for our team. We carefully examined each of your comments and revised the manuscript accordingly, with all changes clearly highlighted in yellow. Below, we provide detailed responses to your points.
>
> **1. Weaknesses 1 — Ablation on global/subgraph/node-level adjacency matrices.**  Thank you very much for this insightful suggestion. As recommended, we have added a **comprehensive ablation study** isolating the three components of our hierarchical correlation graph: global adjacency $\mathcal{A}^{G}$, subgraph (DSS) adjacency $\mathcal{A}^{D}$, node-level adaptive adjacency $\mathcal{A}^{A}$, along with pairwise combinations $\mathcal{A}^{GD}$ and $\mathcal{A}^{DA}$, and the full offline fusion (w/o ol).
> **Summary of the new offline ablation (METR-LA) (MAE only for brevity; full metrics in Section 3.2, Lines 432-461 (Table 2 & Fig.3)):**
>
> | Horizon | Metric | $\mathcal{A}^{G}$ | $\mathcal{A}^{D}$ | $\mathcal{A}^{A}$ | $\mathcal{A}^{GD}$ | $\mathcal{A}^{DA}$ | w/o ol |
> |--------|--------|------|------|------|------|------|--------|
> | H-3 | MAE | 2.60 | 2.55 | 2.63 | 2.54 | 2.54 | **2.52** |
> | H-6 | MAE | 3.00 | 2.92 | 3.01 | 2.92 | 2.92 | **2.89** |
> | H-12| MAE | 3.41 | 3.31 | 3.42 | 3.34 | 3.33 | **3.27** |
>
> These results show that $\mathcal{A}^{G}$, $\mathcal{A}^{D}$, and $\mathcal{A}^{A}$ each bring non-trivial predictive gains. Pairwise combinations consistently yield further improvements, demonstrating that each level captures complementary structural information at different spatial scales. The full fusion variant (w/o ol) achieves the strongest results across all horizons. These findings support that the hierarchical adjacency design is **beneficial rather than redundant**. The new table and discussion have been added to **Section 3.2, Lines 405–461**.
>
> **2. Weaknesses 2 — Hyperparameter robustness analysis.**  Thank you for raising this important point. We have extended our robustness study to include the remaining key hyperparameters of the online adaptation module—**drift-detection window size ($W$)**, **subgraph expansion size ($N_{\mathrm{sg}}^{\max}$)**, and **Top-$K$ candidates for approximate drift-pattern matching**. Across all evaluated ranges, **GAMAD remains stable**, showing only marginal variations in MAE/RMSE/MAPE (see **Appendix F**).
> We also provide clearer rationale in the revision:
> - The **window size ($W$)** controls the temporal granularity for detecting short-term distribution shifts while naturally resisting noise.
> - The **subgraph expansion size ($N_{\mathrm{sg}}^{\max}$)** balances locality vs. expressiveness—too small restricts the drift region, whereas excessively large expansion dilutes the drift signal.
> - The **Top-$K$ matching parameter** determines how many historical patterns are reused. We adopt a normalized linearly decaying scheme $[K, K-1, \dots, 1]$, ensuring that more similar patterns receive higher influence and noisy matches are automatically downweighted.
>
> Full robustness curves and discussions have been added to **Appendix F**, with clarifications in **Lines 855–879 & 918–964 (Fig. 5)**.

---

> ### Author Response · Authors · 2025-11-19
>
> **3. Weaknesses 3 — Computational overhead and scalability.**  We appreciate this valuable concern and have added both **runtime experiments** and a **concise complexity analysis** in the revision. Below we summarize the key results (full details in **Appendix I**).
>
> **Runtime experiments (Lines 999–1013):**
>
> | Method / Setting | Offline Time (s/epoch) | Online Update Latency (s/step) |
> |------------------|-------------------------|--------------------------------|
> | **Ours (GAMAD)** | **64.63**               | **1.27** |
> | CGLM (dynamic graph) | 55.74               | 1.11 |
> | GTS (no predefined graph) | 49.26          | — |
>
> These results indicate that:
> (1) Despite hierarchical graph construction and dynamic correlation updates, GAMAD’s offline runtime remains comparable to dynamic-graph baselines such as CGLM;
> (2) The online update latency of **1.27s** is far smaller than the **5-minute multi-stream updating interval** of METR-LA, satisfying real-time requirements;
> (3) The overall runtime suggests that the proposed graph-driven adaptation mechanism does **not** introduce prohibitive computational overhead.
>
> **Complexity analysis (Lines 1015–1025):**
>
> The global correlation graph is constructed only once with complexity **$O(N^2)$**, while online adaptation operates only on a localized subgraph of size $k < 20$, yielding **$O(k)$** per update. Because updates modify only local correlations, global adjacency matrices do not need to be recomputed. Experiments on METR-LA (207 nodes) and PEMS-BAY (325 nodes) show stable runtime, supporting the scalability of the framework to larger multi-stream systems.
>
> **4. Weaknesses 4 — Statistical significance analysis.**  Thank you for this helpful suggestion. To verify that the improvements of **GAMAD** over **CGLM** are not due to random fluctuations, we conducted **paired two-tailed $t$-tests at the 0.05 significance level** on the METR-LA dataset. Prediction horizons (H-3, H-6, H-12) were treated as paired samples.
>
> | Metric | Mean diff. (CGLM − GAMAD) | t-statistic | p-value |
> |--------|----------------------------|-------------|---------|
> | MAE    | 0.05                       | 8.66        | 0.013   |
> | RMSE   | 0.11                       | 19.05       | 0.0027  |
> | MAPE   | 0.157                      | 9.40        | 0.011   |
>
> All $p$-values are below 0.05, confirming that the gains of GAMAD over CGLM are **statistically significant**. Full details are reported in **Appendix H, Lines 975–992**, marked in yellow.
>
> Thank you again for your careful reading and constructive feedback. Your suggestions helped us significantly refine the analyses, clarify the design decisions, and strengthen the experimental evidence. We are always eager to hear from you. We hope that the revisions and detailed explanations adequately address your concerns, and we would be truly grateful if you could consider raising your score in light of the improvements.

---

### Author Response · Authors · 2025-11-29
**Review and Rebuttal Summary for Area Chair**

Dear Area Chair,

Thank you for evaluating our submission under the updated review process.
Below is a concise summary of the **main reviewer concerns** and the **new experiments and clarifications** added during rebuttal.
This aims to help you quickly understand the state of the review and rebuttal discussion before it was frozen.

## 1. Consolidated Reviewer Concerns
*(All reviewer indices follow page order: R1, R2, R3, R4)*

All reviewers agree that the paper tackles an **important and realistic problem** in multi-stream environments with concept drift and correlated streams.

They highlight that:
- The **motivation and problem setting** are strongly grounded in real-world traffic and weather applications (R1–R4).
- The method **removes reliance on predefined topology** through dynamic correlation learning (R2–R4).
- GAMAD provides a **complete, graph-driven adaptation pipeline**: node-level detection, localized updates, and drift-pattern reuse (R3, R4; also noted by R1, R2).
- Experiments show **consistent and substantial improvements** across multiple datasets (R1–R4).

Reviewers raised several focused concerns that guided our revisions:

- **Ablation completeness** — R1–R4
- **Hyperparameter robustness** — R1, R3
- **Computational overhead & scalability** — R1–R3
- **Noise robustness & drift-detection reliability** — R2, R3
- **Originality clarification** — R2
- **Behavior under rapidly changing correlations** — R2, R3
- **Statistical significance of gains** — R1

## 2. Summary of Rebuttal and Newly Added Evidence
*(full details in rebuttal and yellow-highlighted manuscript)*

### (A) Complete Offline + Online Ablation

We added a comprehensive two-stage ablation:

- **Offline:** $\mathcal{A}^{G}$, $\mathcal{A}^{D}$, $\mathcal{A}^{A}$, their pairwise fusions, and full fusion.
- **Online:** naïve online learning → +drift detection → +dynamic subgraph expansion → +pattern reuse.

Outcome: Each adjacency scale and each online module contributes meaningfully. Fused variants consistently outperform single components, and the full model achieves the best performance.
**Addresses concerns from R1–R4.**

### (B) Hyperparameter Robustness

We extended robustness tests to three online-critical parameters: drift-detection window size $W$, maximum subgraph expansion $N_{\mathrm{sg}}^{\max}$ and top-$K$ pattern candidates.

Outcome: Performance curves remain stable across wide ranges.
**Addresses concerns from R1, R3**

### (C) Runtime and Complexity Analysis

We added related measurements and discussion: GAMAD’s **offline runtime** is comparable to dynamic-graph baselines. **Online latency** (~1.27s) is far below the 5-minute sampling interval → **real-time safe**.  **Complexity** is $O(N^2)$ (one time only) for offline correlation construction and $O(k)$, $k<20$ for online (per update).

**Addresses concerns from R1–R3.**

### (D) Noise Robustness Stress Test

We injected: Gaussian noise (**$0.1 \cdot \mathrm{std}$**), spike perturbations (**2% nodes ×1.5×**) into online inputs.

Outcome: MAE/RMSE/MAPE change **<1%**. Drift ratio does not increase; the detector becomes slightly more conservative.
**Addresses concerns from R2, R3.**

### (E) Statistical Significance

We conducted paired two-tailed **$t$-tests (0.05 level)** on METR-LA dataset.

Outcome: All metrics (MAE/RMSE/MAPE) yield **p < 0.05**, confirming gains over CGLM are statistically meaningful.
**Addresses concerns from R1.**

### (F) Clarified Conceptual Novelty

We clarified that GAMAD provides a **structure-aware, graph-driven adaptation framework**, including:

1. hierarchical correlation graph learning without predefined topology;
2. real-time dynamic subgraph adaptation centered on drift nodes;
3. hierarchical drift-pattern reuse across streams.

**Addresses concerns from R2.**

### (G) Handling Rapidly Changing Correlations

We emphasized that dynamic subgraphs are **reconstructed at every drift step using current correlations**, enabling fast response without recomputing global graphs. The noise stress test further supports robustness under abrupt correlation changes.
**Addresses concerns from R2, R3.**

## 3. Reviewer Reception After Rebuttal

Before freeze, reviewer **R4** (initially 4/10) responded:

> “The rebuttal and additional experiments address some of my concerns… I remain positive about this paper and will increase my rating accordingly.”

Given overlapping concerns, this strongly suggests the added evidence satisfactorily resolved major issues.

---

Reviewers consistently recognized the **importance**, **practical relevance**, and **empirical strength** of our work.
We substantially improved the manuscript through extensive new experiments, robustness analyses, scalability evaluation, and clearer conceptual positioning.

We respectfully ask the Area Chair to consider these improvements and the positive reviewer response when forming the meta-review.

Thank you very much for your time and consideration.

---

### Meta-Review · Area_Chair_TxjN · 2026-01-01

**Summary:**

The reviewers raised several major concerns. First, the novelty of the proposed method was questioned, as its core motivation and solution largely consist of a combination of existing and well‑studied techniques. Second, the initial submission lacked sufficient ablation studies to analyze the contributions of the proposed components. Third, the computational complexity of the method may be high. Fourth, the behavior of the method under special scenarios, such as highly dynamic flows and robustness to noise perturbations, has not been sufficiently explored. Finally, the performance improvements over existing baselines are relatively limited.

I believe that during the rebuttal phase, the authors addressed some of these concerns by providing additional ablation experiments, robustness experiments and statistical significance analyses, which help demonstrate certain performance advantages of the proposed approach. However, although the added experiments indicate reasonable scalability, the method incurs noticeably higher runtime costs than the baselines while achieving only modest performance gains, and its computational complexity(O(n^2) is not ideal) does not appear suitable for very large graphs. Moreover, I find the authors’ explanation regarding novelty insufficiently convincing, as the overall level of innovation remains limited. Finally, the presentation could be improved. For example, Fig. 2 are excessively large and detract from the visual quality of the paper.

For these reasons, I recommend rejecting the paper.

**Reviewer Concerns:**

Ablation studies.
For the concern regarding the lack of ablation studies, I believe the authors addressed this issue by providing additional ablation experiments in the rebuttal, which clarify the contributions of the individual components and partially strengthen the experimental support.


Robustness and noise perturbations.
For the concern regarding robustness under noise perturbations, I believe the authors conducted additional robustness experiments, which demonstrate a certain degree of stability of the proposed method under perturbations.

Statistical significance.
Regarding the statistical significance of the reported improvements, I believe the author provided a statistical significance analysis in the rebuttal, which helps to illustrate the effectiveness of the method's enhancements

Computational complexity and scalability.
Considering the computational complexity and scalability, I think the experiments added by the author are not convincing enough. When the experimental effect is slightly improved, the additional computational consumption of the model is relatively large. In addition, theoretical complexity still poses challenges on extremely large graphs.

Novelty.
For the concern regarding novelty, the authors attempted to further justify the novelty of the proposed method in the rebuttal;  however, this explanation is not sufficiently convincing, and the overall level of innovation remains limited.

**Reviewer Scores:**

For Reviewer Jqa6 and m19R, I think they will keep their positive scores unchanged.
For Reviewer 7CVj, I think he will maintain his clear refusal and keep the score at 2 unchanged.
For Reviewer 7YEb, I think he will raise his score to 6.

---

### Decision · Program_Chairs · 2026-01-26

Reject